# UNDERSTANDING AND SCHEDULING WEIGHT DECAY

## ABSTRACT

Weight decay is a popular and even necessary regularization technique for training deep neural networks that generalize well. Previous work usually interpreted weight decay as a Gaussian prior from the Bayesian perspective. However, weight decay sometimes shows mysterious behaviors beyond the conventional understanding. For example, the optimal weight decay value tends to be zero given long enough training time. Moreover, existing work typically failed to recognize the importance of scheduling weight decay during training. Our work aims at theoretically understanding novel behaviors of weight decay and designing schedulers for weight decay in deep learning. This paper mainly has three contributions. First, we propose a novel theoretical interpretation of weight decay from the perspective of learning dynamics. Second, we propose a novel weight-decay linear scaling rule for large-batch training that proportionally increases weight decay rather than the learning rate as the batch size increases. Third, we provide an effective learning-rate-aware scheduler for weight decay, called the Stable Weight Decay (SWD) method, which, to the best of our knowledge, is the first practical design for weight decay scheduling. In our various experiments, the SWD method often makes improvements over $L_2$ Regularization and Decoupled Weight Decay.

## 1 INTRODUCTION

Weight decay is a popular and even necessary regularization technique for training deep neural networks that generalize well (Krogh and Hertz, 1992). People commonly use $L_2$ regularization as "weight decay" for training of deep neural networks and interpret it as a Gaussian prior over the model weights (David, 1992; Graves, 2011). This is true for vanilla Stochastic Gradient Descent (SGD). However, Loshchilov and Hutter (2018) revealed that, when the learning rate is adaptive, the commonly used $L_2$ regularization is not identical to the vanilla weight decay proposed by Hanson and Pratt (1989):

$$\theta_t = (1 - \lambda')\theta_{t-1} - \eta\frac{\partial L(\theta_{t-1})}{\partial \theta}, \tag{1}$$

where $\lambda'$ is the weight decay hyperparameter, $\eta$ is the learning rate, and $L(\theta_{t-1})$ is the loss function over one minibatch. We denote the loss function over the whole training dataset as $L(\theta)$.

A conventional belief is that, from the Bayesian perspective, weight decay indicates a Gaussian prior over the model parameters, which regularize the complexity of the learned model (Graves, 2011). However, we argue that this Bayesian interpretation cannot explain the empirical success of weight decay in deep learning. On one hand, Wenzel et al. (2020) demonstrated that the conventional Bayesian posterior usually does not work well in deep learning. This challenged the meaning of weight decay as a proper prior in the Bayesian interpretation. On the other hand, Lewkowycz and Gur-Ari (2020) and our experimental results (Figure 2) both suggested that weight decay is surprisingly unnecessary for long-time training. This observation critically challenged the Bayesian interpretation of weight decay in deep learning. The Bayesian interpretation believes that the prior knowledge over model parameters truly exist. Thus, the prior knowledge should not depend on the training time of deep neural networks, which contradicts with the experimental results in Figure 2.

Recently researchers started to explore other explanations for weight decay. Zhang et al. (2019b) revealed three different roles of weight decay. Some papers (Van Laarhoven, 2017; Zhang et al., 2019b) argued that weigh decay increases the effective learning rate but has no regularization effect

Table 1: Test performance comparison of optimizers. We report the mean and the standard deviations (as the subscripts) of the optimal test errors computed over three runs of each experiment. The SWD method enables Adam to generalize as well as SGD and even outperform complex Adam variants.

| DATASET | MODEL | SGD | ADAMS | ADAM | AMSGRAD | ADAMW | ADABOUND | PADAM | YOGI | RADAM |
|---------|-------|-----|-------|------|---------|-------|----------|-------|------|-------|
| CIFAR-10 | RESNET18 | $5.01_{0.03}$ | $\mathbf{4.91}_{0.04}$ | $6.96_{0.02}$ | $6.16_{0.18}$ | $5.08_{0.07}$ | $5.65_{0.08}$ | $5.12_{0.04}$ | $5.87_{0.12}$ | $6.01_{0.10}$ |
| | VGG16 | $6.42_{0.02}$ | $\mathbf{6.09}_{0.11}$ | $7.31_{0.25}$ | $7.14_{0.14}$ | $6.59_{0.13}$ | $6.76_{0.12}$ | $6.15_{0.06}$ | $6.90_{0.22}$ | $6.56_{0.04}$ |
| CIFAR-100 | RESNET34 | $\mathbf{21.52}_{0.37}$ | $21.76_{0.42}$ | $27.16_{0.55}$ | $25.53_{0.19}$ | $22.99_{0.40}$ | $22.87_{0.13}$ | $22.72_{0.10}$ | $23.57_{0.12}$ | $24.41_{0.40}$ |
| | DENSENET121 | $\mathbf{19.81}_{0.33}$ | $20.52_{0.26}$ | $25.11_{0.15}$ | $24.43_{0.09}$ | $21.55_{0.14}$ | $22.69_{0.15}$ | $21.10_{0.23}$ | $22.15_{0.36}$ | $22.27_{0.22}$ |
| | GOOGLENET | $21.21_{0.29}$ | $\mathbf{21.05}_{0.18}$ | $26.12_{0.33}$ | $25.53_{0.17}$ | $21.29_{0.17}$ | $23.18_{0.31}$ | $21.82_{0.17}$ | $24.24_{0.16}$ | $22.23_{0.15}$ |

when combined with scale-invariant normalization layers, such as batch normalization (Ioffe and Szegedy, 2015). Li et al. (2020) defined a new "intrinsic learning rate" parameter that is the product of the vanilla learning rate $\eta$ and the weight decay factor $\lambda$ and argued that the convergence time depends on this intrinsic learning rate.

However, people still expect to further explore the theoretical mechanism of weight decay, because weight decay sometimes shows mysterious behaviors beyond the existing theoretical understanding. As we discussed above, the Bayesian interpretation has been critically challenged by recent empirical findings (Lewkowycz and Gur-Ari, 2020). The interpretation of increasing the effective learning rate only holds for scale-invariant normalization layers, which is a narrow scope.

**Contributions.** In this paper, we try to theoretically understand the novel behaviors of weight decay and explore the potential of weight decay. Our paper mainly has three contributions.

First, we propose a novel theoretical interpretation of weight decay from the perspective of learning dynamics. We study how the stability of the stationary points and the convergence time depend on weight decay, which corresponds to two main effects of weight decay.

Second, we propose a novel weight-decay linear scaling rule (See Rule 2 in Section 3) for large-batch training. We discover that the existing linear scaling rule (Krizhevsky, 2014; Goyal et al., 2017) for large-batch training that proportionally increases the learning rate may sometimes be harmful to large-batch training due to bad convergence. Instead, increasing weight decay is a more suitable solution in this case.

Third, to the best of our knowledge, we are the first to design an effective learning-rate-aware scheduler for weight decay which may significantly improve test performance. Our work fills the gap in this direction. We note that a trivial weight decay scheduling method recently proposed by Lewkowycz and Gur-Ari (2020) has limited practical value because it only works well with constant learning rate, as the original paper claimed. Inspired by the stability of the stationary points, we show that weight decay should be coupled with the effective learning rate. We call the proposed method Stable Weight Decay (SWD). Adam (Kingma and Ba, 2015) with SWD (AdamS) is displayed in Algorithm 2. The results in Table 1 fully supports that SWD can significantly improve Adam.

**Structure of this paper.** In Section 2, we analyze the dynamics of weight decay. In Section 3, we study how weight decay can improve large-batch training. In Section 4, we propose SWD. In Section 5, we empirically study SWD. In Section 6, we conclude our work.

## 2 UNDERSTANDING THE DYNAMICS OF WEIGHT DECAY

In this section, we provide a way to understand weight decay from a viewpoint of learning dynamics.

**Weight decay should be coupled with the learning rate scheduler.** The vanilla weight decay described by Hanson and Pratt (1989) is given by Equation (1). A more popular implementation for vanilla SGD in modern deep learning libraries, such as PyTorch (Paszke et al., 2019) and TensorFlow (Abadi et al., 2016), is given by

$$\theta_t = (1 - \eta_t \lambda)\theta_{t-1} - \eta_t \frac{\partial L(\theta_{t-1})}{\partial \theta}, \tag{2}$$

where $\eta_t$ is the learning rate at the $t$-th step and weight decay is coupled with the learning rate scheduler. While previous work did not studied why weight decay should be coupled with the learning rate scheduler, this trick has been adopted by most deep learning libraries.

We first take vanilla SGD as the studied example, where no Momentum is involved. It is easy to see that vanilla SGD with $L_2$ regularization $\frac{\lambda}{2}\|\theta\|^2$ is also given by Equation (2). Suppose the learning rate is fixed in the whole training procedure. Then Equation (2) will be identical to Equation (1) if we simply choose $\lambda' = \eta\lambda$. However, learning rate decay is quite important for training of deep neural networks. So Equation (2) is not identical to Equation (1) in practice.

Which implementation is better? In Figure 1, we empirically verified that the popular Equation-(2)-based weight decay indeed outperforms the vanilla implementation in Equation (1). We argue that Equation-(2)-based weight decay is theoretically better than Equation-(1)-based weight decay in terms of the stability of the stationary points. We define the stability as Definition 1.

**Definition 1** (The stability of the stationary point). *The stationary point is stable if a stationary point is fixed during training. Otherwise, we define it as an unstable stationary point.*

The training objective at step-$t$ of Equation (1) is $L(\theta) + \frac{\lambda'}{\eta_t}\|\theta\|^2$, while that of Equation (2) is $L(\theta) + \lambda\|\theta\|^2$. Thus, the stationary points of the regularized loss given by Equation (2) are stable during training, while that given by Equation (1) is unstable due to the dynamic training objective.

**Weight decay matters to the stability.** As the stability of the stationary points critically depends on the weight decay scheduler, we define stable weight decay accordingly in Definition 2.

**Definition 2** (Stable Weight Decay). *The weight decay is stable if the stationary points are stable during training. Otherwise, we define it as unstable weight decay.*

We propose Theorem 1 and prove that even deterministic Gradient Descent (GD) with vanilla weight decay may not converge to any non-zero stationary point due the instability of the stationary points.

**Theorem 1** (Non-convergence of GD with unstable weight decay). *Suppose learning dynamics is governed by GD with vanilla weight decay (Equation (1)) and the learning rate $\eta_t \in (0, +\infty)$ holds. If $\exists \delta$ that satisfies $0 < \delta \leq |\eta_t - \eta_{t+1}|$ for any $t > t_0$, then the learning dynamics cannot converge to any non-zero stationary point satisfying the condition*

$$\lim_{t \to +\infty} [\|\nabla L(\theta_t)\|^2 + \|\nabla L(\theta_{t+1})\|^2] = 0.$$

**Remark.** *We leave the proof in Appendix A.1. Even if the gradient is zero at the $t$-th step, the gradient at the $t + 1$-th step will not be zero. Unstable weight decay causes changing the objective and hence the solution does not converge to any non-zero stationary point. Thus, GD with unstable weight decay has no theoretical convergence guarantee like GD (Wolfe, 1969). This explains why weight decay should be coupled with the learning rate to keep the stationary points stable during training. We conjecture that the stability of the stationary points should be a desirable property in deep learning.*

**Dynamics of weight decay.** Lewkowycz and Gur-Ari (2020) empirically discovered that, if we let the training time be approximately inverse to weight decay, the test error can monotonically decrease as weight decay decreases, which is beyond the existing theoretical understanding. We will show that it is easy to understand this empirical observation by analyzing the dynamics of weight decay.

We first state Assumption 1. Assumption 1 which holds near minima are common and useful for analyzing the solution or the convergence behavior in related papers (Mandt et al., 2017; Neyshabur et al., 2017; Xie et al., 2021a; Li et al., 2017; Zhang et al., 2019a; Zhou et al., 2020). It is known that stochastic optimization can escape multiple bad minima (Xie et al., 2021b; Zhu et al., 2019; Kleinberg et al., 2018) and finally converge to a good minimum during training. Thus, Assumption 1 can be repeatedly applied near each minimum during training.

**Assumption 1.** *The loss function around a minimum $\theta^\star$ can be written as*

$$L(\theta) = L(\theta^\star) + \frac{1}{2}(\theta - \theta^\star)^\top H(\theta - \theta^\star),$$

*where $H$ is the Hessian at $\theta^\star$.*

We propose Theorem 2 and demonstrate how the learned model parameters $\theta$ may depend on weight decay. We leave the proof in Appendix A.2.

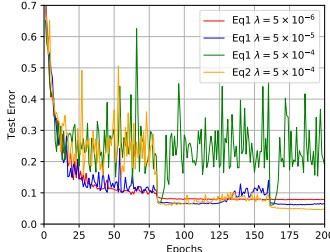 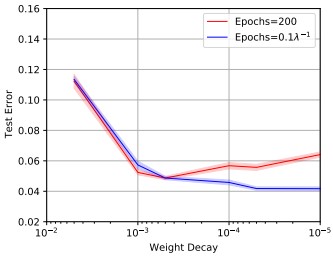

Figure 1: We compared Equation-(1)-based weight decay and Equation-(2)-based weight decay by training ResNet18 on CIFAR-10 via vanilla SGD. In the presence of a popular learning rate scheduler, Equation-(2)-based weight decay shows better test performance. It demonstrates that the form $-\eta_t \lambda \theta$ is a better weight decay implementation than $-\lambda'\theta$.

Figure 2: We train ResNet18 via SGD on CIFAR-10 for verifying that $t_{\text{convergence}} = \mathcal{O}\left(\lambda^{-1}\right)$. With the fixed 200 epochs, the optimal weight decay is about 0.0005. With $0.1\lambda^{-1}$ epochs, decreasing weight decay monotonically decreases test performance. Table 3 further supports that the optimal weight decay is approximately inverse to the number of epochs.

**Theorem 2** (Dynamics of weight decay). *Suppose Assumption 1 holds, and learning dynamics is governed by SGD with weight decay (Equation* (2)*). If the learning rate scheduler is constant, the expected $\theta_t$ after $t$ iterations satisfies the following:*

$$\mathbb{E}[\theta_t - \theta^\star] = [I - \eta(H + \lambda I)]^t \left[\theta_0 - H(H + \lambda I)^{-1}\theta^\star\right] - \lambda(H + \lambda I)^{-1}\theta^\star, \tag{3}$$

*where $I$ is the identity matrix. If the learning rate scheduler is given by the sequence $\{\eta_1, \eta_2, \ldots, \eta_t\}$, the expected $\theta_t$ after $t$ iterations satisfies the following:*

$$\mathbb{E}[\theta_t - \theta^\star] = \prod_{k=1}^{t}[I - \eta_k(H + \lambda I)]\left[\theta_0 - H(H + \lambda I)^{-1}\theta^\star\right] - \lambda(H + \lambda I)^{-1}\theta^\star. \tag{4}$$

Theorem 2 is a useful tool to understand convergence behaviors which may repeatedly happen during searching minima of deep learning. While Theorem 2 depends on Assumption 1, we will show that it is remarkably consistent with novel empirical behaviors of weight decay and can also reveal novel insights. Theorem 2 indicates that weight decay mainly has two effects: 1) biasing the expected learned solution and 2) accelerating the convergence.

**Effect 1. Weight decay biases the stationary points.** Theorem 2 provides us two key insights about weight decay. The first insight is Corollary 1 derived from Theorem 2 by choosing the long-time limit. Corollary 1 indicates that large weight decay may not learn accurate minima well. This insight is consistent with the regularization-based understanding of weight decay (Krogh and Hertz, 1992).

**Corollary 1.** *Suppose the conditions of Theorem 2 hold and $H_i + \lambda > 0$, where $H_i$ is the $i$-th eigenvalue of $H$. Then, in the limit of long training time that $t \to +\infty$, the displacement from the true minimum $\theta^\star$ to the expected learned solution along the $i$-th direction is given by*

$$\mu_i = -\lambda(H_i + \lambda)^{-1}\theta^\star, \tag{5}$$

*where the bias $\mu = \lim_{t\to\infty} \mathbb{E}[\theta_t - \theta^\star]$.*

Figure 2 suggests that, given long enough training time inverse to weight decay, decreasing the weight decay hyperparameter may consistently improve test performance, which is also verified by Lewkowycz and Gur-Ari (2020). This observation is contradicted with the conventional regularization-based understanding (Krogh and Hertz, 1992), but fully supports Corollary 1 that weight decay is bad, if we already have good convergence.

**Effect 2. Weight decay accelerates convergence.** However, we discover that the long-time posterior cannot explain the convergence behaviors. Figure 2 also suggests that, we need to increase the number of training iterations by $\frac{1}{\lambda}$ for maintaining good convergence, which is also discovered by Lewkowycz and Gur-Ari (2020) and Li et al. (2020). To explain why the convergence time depends on $\frac{1}{\lambda}$, we need to explore the dynamics of weight decay before approaching the convergence.

The second insight is Corollary 2 that estimates the number of iterations for good convergence.

**Corollary 2.** *Suppose the conditions of Theorem 2 hold and $\delta_i = H_i + \lambda > 0$, where $H_i$ is the $i$-th eigenvalue of $H$. Then, the convergence time in the $i$-th direction is*

$$t_{\text{convergence},i} = \mathcal{O}\left((H_i + \lambda)^{-1}\eta^{-1}\right). \tag{6}$$

It is well-known that the Hessian in deep learning has a very small number of large eigenvalues and a large number of nearly zero eigenvalues (Sagun et al., 2016; 2017). Thus, we have $H_i + \lambda \approx \lambda$ for most directions. It means that the convergence time critically depend on the dynamics in the subspace corresponding to those nearly zero eigenvalues:

$$t_{\text{convergence}} = \mathcal{O}\left(\lambda^{-1}\eta^{-1}\right). \tag{7}$$

Equation (7) reveals two facts. First, Equation (7) supports the statement that the initial learning rate and the weight decay hyperparameter $\lambda$ are linearly coupled for maintaining performance (Loshchilov and Hutter, 2018; Van Laarhoven, 2017; Zhang et al., 2019b; Hoffer et al., 2018). Second, Equation (7) demonstrates that the number of iterations (not the number of epochs or training time) for good convergence depends on $\mathcal{O}\left(\lambda^{-1}\eta^{-1}\right)$. This not only explains the relation in Figure 2, but also reveals that we should increase the number of iterations instead of the number of epochs by $\frac{1}{\lambda}$. We note that the second fact is also reported by Lewkowycz and Gur-Ari (2020), while Lewkowycz and Gur-Ari (2020) did not present formal theoretical analysis like Theorem 2.

In our theoretical analysis, weight decay mainly has two effects. The first effect is to bias the expected solution in the long-time limit. The first effect is negative for learning accurate minima. The second effect is to accelerate the convergence in the subspace corresponding to nearly zero eigenvalues of the Hessian. The second effect is positive for accelerating training. Tuning weight decay needs to balance the tradeoff between the two effects.

## 3 WEIGHT DECAY IMPROVES LARGE-BATCH TRAINING

In this section, motivated by the theoretical insights in Section 2, we study how weight decay may surprisingly improve large-batch training.

Large-batch training can efficiently utilize the parallel computation to speedup training of deep networks (Goyal et al., 2017). However, large-batch training suffers from the difficulty of minimizing the training loss and usually find sharp minima that do not generalize well (Hoffer et al., 2017; Keskar et al., 2017). How to perform large-batch training with good generalization is an important problem which has attracted much attention from both academia (Zhang et al., 2019a; Wu et al., 2020; Wen et al., 2020) and industry (You et al., 2017; Goyal et al., 2017).

**Rule 1 mitigates the small noise problem in large-batch training.** The famous linear scaling rule, namely Rule 1, for large-batch training is well-known (Krizhevsky, 2014; Goyal et al., 2017). Large-batch training can easily get stuck in saddle points and sharp minima due to small gradient noise, because the noise magnitude in SGD dynamics is approximately proportional to the ratio of the learning rate $\eta$ to batch size $B$, namely $\frac{\eta}{B}$ (Mandt et al., 2017; Keskar et al., 2017). Xie et al. (2021b) recently proved that $\frac{\eta}{B}$ *exponentially* matters to SGD dynamics. Thus, the learning-rate linear scaling rule is a natural result of maintaining the similar magnitude of stochastic gradient noise and similar learning dynamics invariant to the batch size. When the noise magnitude is the performance bottleneck, Rule 1 is usually an effective solution.

**Rule 1.** *When the batch size is multiplied by $k$, multiply the learning rate by $k$.*

For large-batch training, people usually focused on the noise magnitude problem but overlooked the convergence problem (Jain et al., 2018; Ma et al., 2018; Yin et al., 2018). Sometimes, the noise magnitude is not the performance bottleneck. People usually perform large-batch training with a fixed number of epochs for keeping the total computational costs moderate. Under the constraint of fixing the number of epochs, obviously, the number of iterations will be multiplied by $k^{-1}$ if the batch size is multiplied by $k$. Thus, the number of iterations may be too small to achieve good convergence.

**Rule 2 mitigates the bad convergence problem in large-batch training.** If the bad convergence problem is the performance bottleneck, the learning-rate linear scaling rule will even be harmful to large-batch training due to slower convergence. When Rule 1 causes optimization divergence or bad convergence, we instead propose the *weight-decay linear scaling rule* as Rule 2 based on Equation

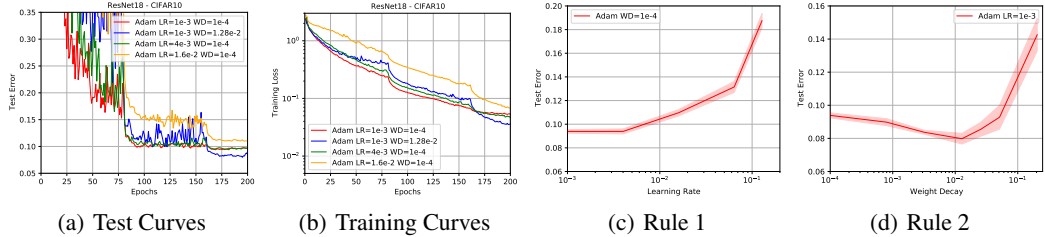

| (a) Test Curves | (b) Training Curves | (c) Rule 1 | (d) Rule 2 |

Figure 3: Large-batch training ($B = 16384$) with various learning rates and weight decay. Note that $\eta = 10^{-3}$ and $\lambda = 10^{-4}$ is the baseline choice for $B = 128$. Subfigure (a) and (b) show that, even slightly increasing the learning rate (by multiplying 16) is harmful to optimization convergence. Subfigure (c) shows that Rule 1 is completely invalid in this common large-batch training setting. Subfigure (d) shows that, multiplying weight decay by 128 ($\lambda = 0.0128$) has the lowest test error, which fully supports the proposed Rule 2.

(7). Because large weight decay may accelerate the convergence. Note that, increasing the learning rate only works in a narrow range, i.e., $B \leq 8192$ on ImageNet (Keskar et al., 2017), since too large learning rates may lead to optimization divergence or bad convergence (Keskar et al., 2017; Masters and Luschi, 2018).To the best of our knowledge, we are the first to propose the weight-decay-based rule for large-batch training. Previous papers did not design a rule for the second-type problem.

**Rule 2.** *When the batch size is multiplied by $k$, multiply the weight decay by $k$.*

**Empirical Analysis of Large-Batch Training.** We conducted experiments to show that, when the learning-rate linear scaling rule is harmful due to the convergence problem, the weight-decay linear scaling rule significantly improves large-batch training. In Figure 3, we trained ResNet18 via SGD on CIFAR-10 with the very large batch size $B = 16384$, which is 128 times the common batch size 128.

We note that, in this common setting on CIFAR-10, even slightly increasing the learning rate is very harmful to large-batch training. Figure 3 demonstrates Rule 1 may seriously cause bad convergence. This observation means that, in the setting of training ResNet18 on CIFAR-10, the convergence problem is the main performance bottleneck instead of the noise magnitude. As Rule 2, we observed that multiplying the weight decay hyperparameter by 128 works best. Thus, in this second-type problem of large-batch training, Rule 2 significantly outperforms Rule 1.

## 4 STABLE WEIGHT DECAY

In this section, we design a weight decay scheduler for adaptive gradient methods, such as Adam.

| **Algorithm 1:** Adam/AdamW | **Algorithm 2:** AdamS ($p = 0.5$) |
|---|---|
| $g_t = \nabla L(\theta_{t-1}) + \lambda\theta_{t-1}$; | $g_t = \nabla L(\theta_{t-1})$; |
| $m_t = \beta_1 m_{t-1} + (1-\beta_1)g_t$; | $m_t = \beta_1 m_{t-1} + (1-\beta_1)g_t$; |
| $v_t = \beta_2 v_{t-1} + (1-\beta_2)g_t^2$; | $v_t = \beta_2 v_{t-1} + (1-\beta_2)g_t^2$; |
| $\hat{m}_t = \frac{m_t}{1-\beta_1^t}$; | $\hat{m}_t = \frac{m_t}{1-\beta_1^t}$; |
| $\hat{v}_t = \frac{v_t}{1-\beta_2^t}$; | $\hat{v}_t = \frac{v_t}{1-\beta_2^t}$; |
| $\theta_t = \theta_{t-1} - \frac{\eta}{\sqrt{\hat{v}_t}+\epsilon}\hat{m}_t - \eta\lambda\theta_{t-1}$; | $\bar{v}_t = mean(\hat{v}_t)$; |
| | $\theta_t = \theta_{t-1} - \frac{\eta}{\sqrt{\hat{v}_t}+\epsilon}\hat{m}_t - \frac{\eta}{\bar{v}_t^p}\lambda\theta_{t-1}$; |

**Decoupled weight decay is unstable weight decay in adaptive gradient methods.** Loshchilov and Hutter (2018) first pointed that, when the learning rate is adaptive, the commonly used $L_2$ regularization is not identical to weight decay. In the following analysis, we ignore the effect of Momentum and focus on the effect of Adaptive Learning Rate for simplicity. Thus, AdamW (Loshchilov and Hutter, 2018) can be written as

$$\theta_t = (1 - \eta\lambda)\theta_{t-1} - \eta v_t^{-\frac{1}{2}}\frac{\partial L(\theta_{t-1})}{\partial \theta}, \tag{8}$$

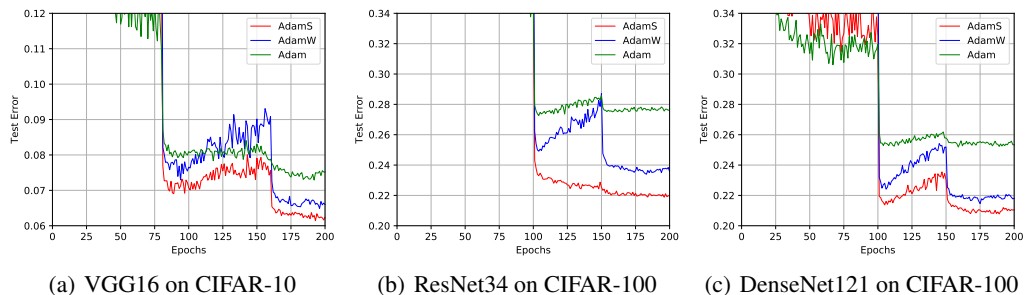

(a) VGG16 on CIFAR-10   (b) ResNet34 on CIFAR-100   (c) DenseNet121 on CIFAR-100

Figure 4: The learning curves of AdamS, AdamW, and Adam on CIFAR-10 and CIFAR-100. AdamS shows significantly better generalization than AdamW and Adam.

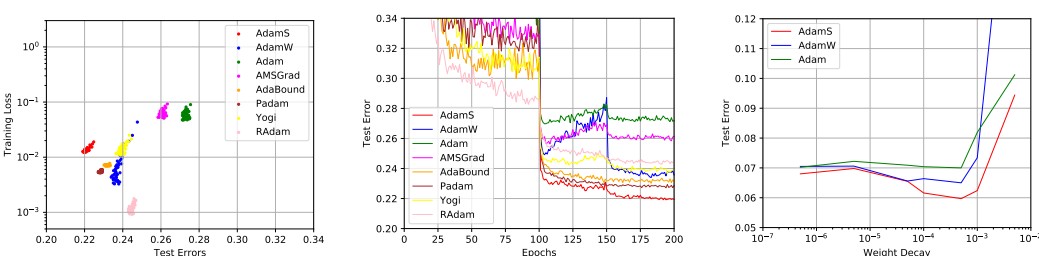

Figure 5: The scatter plot of training losses and test errors during final 40 epochs of training ResNet34 on CIFAR-100. Even with similar or higher training losses, AdamS still generalizes better than other Adam variants. We leave the scatter plot on CIFAR-10 in Appendix C.

Figure 6: The learning curves of all adaptive gradient methods by training ResNet34 on CIFAR-100. AdamS outperforms other Adam variants. The test performance of other models can be found in Table 1.

Figure 7: The test errors of VGG16 on CIFAR-10 with various weight decay rates. The displayed weight decay value of AdamW has been rescaled by the factor $\approx 0.001$. A similar experimental result for ResNet34 is presented in Appendix C.

where $v_t$ is the exponential moving average of the squared gradients in Algorithm 1 and the power notation of a vector means the element-wise power of the vector. We interpret $\eta v_t^{-\frac{1}{2}}$ as the effective learning rate for multiplying the gradients. However, we clearly see that decoupled weight decay uses the vanilla learning rate to perform weight decay rather than the effective learning rate. The minimum $\hat{\theta}^\star$ of the regularized loss function optimized by AdamW at the $t$-th step is

$$\hat{\theta}^\star - \theta^\star = -\lambda v_t^{\frac{1}{2}} (H + \lambda v_t^{\frac{1}{2}})^{-1} \theta^\star. \tag{9}$$

There is no guarantee that $v_t$ will be constant for $t$. Thus, the regularized loss function optimized by AdamW has unstable minima.

**Scheduling weight decay in adaptive gradient methods.** An easy solution to fix the unstable weight decay problem is using the following scheduled weight decay:

$$\theta_t = (\mathbf{1} - \eta v_t^{-\frac{1}{2}} \lambda) \theta_{t-1} - \eta v_t^{-\frac{1}{2}} \frac{\partial L(\theta_{t-1})}{\partial \theta}. \tag{10}$$

This gives $\hat{\theta}^\star - \theta^\star = -\lambda (H + \lambda)^{-1} \theta^\star$. Equation (10) can indeed make stationary points of the regularized loss function stable for different $t$.

However, if we still use the element-wise scheduler $v_t^{-\frac{1}{2}}$ for weight decay, the anisotropic convergence problem will appear in the subspace corresponding to small eigenvalues of the Hessian. By Theorem 2, the convergence can be measured by $\prod_{k=1}^{t}(I - v_k^{-\frac{1}{2}} \eta \lambda)$. Due to the element-wise learning rate adaptivity, a common number of iterations may be not enough for achieving the convergence along some dimensions.

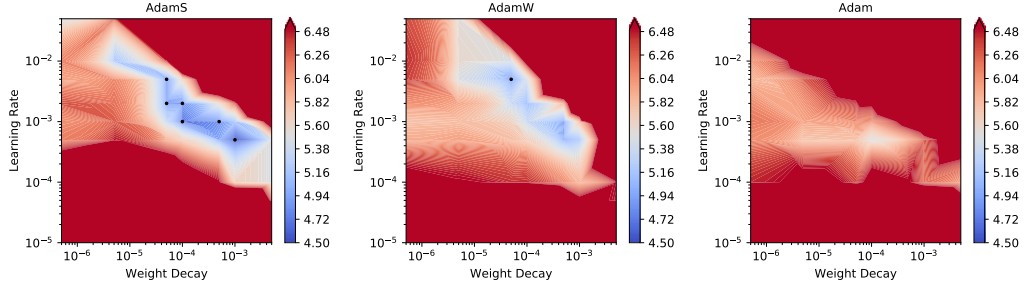

Figure 8: The test errors of ResNet18 on CIFAR-10. AdamS has a much deeper and wider basin near dark points ($\leq 4.9\%$). The optimal test error of AdamS, AdamW, and Adam are $4.52\%$, $4.90\%$, and $5.49\%$, respectively. The displayed weight decay value of AdamW has been rescaled by the factor $\approx 0.001$.

Unfortunately, there is no ideal solution to fix the stability problem and the convergence problem at the same time for adaptive gradient methods. We conjecture that this may be an internal fault of all optimizers that use Adaptive Learning Rate. To balance the tradeoff between the stability and the convergence in presence of Adaptive Learning Rate, we propose a non-element-wise learning-rate-aware scheduler for weight decay as

$$\theta_t = (\mathbf{1} - \eta\bar{v}_t^{-p}\lambda)\theta_{t-1} - \eta v_t^{-\frac{1}{2}}\frac{\partial L(\theta_{t-1})}{\partial\theta}, \tag{11}$$

where $\bar{v}_t$ is the mean of all elements of the vector $v_t$ and $p$ is a hyperparameter to control the adaptivity of the scheduler. Note that $\eta\bar{v}_t^{-p}$ is always isotropic. We call weight decay with this stability-inspired heuristic scheduler Stable Weight Decay. The pseudocode of AdamS is displayed in Algorithm 2.

We recommend $p = 0.5$ as the default value in AdamS unless we specify it, because $\eta\bar{v}_t^{-p}$ with $p = 0.5$ may approximate the magnitude of the adaptive learning rate. Note that Decoupled Weight Decay (Loshchilov and Hutter, 2018; Bjorck et al., 2021) is a special case of Stable Weight Decay with $p = 0$.

We note that $\bar{v}$ is not expected to be zero at minima due to stochastic gradient noise, because the variance of the stochastic gradient is directly observed to be much larger than the expectation of the stochastic gradient at/near minima (Zhu et al., 2019) and depends on the Hessian (Xie et al., 2021b). In some special cases, such as full-batch training, it is fine to add a small value (i.e., $10^{-8}$) to $\bar{v}$ to avoid being zero as a divisor.

## 5 EMPIRICAL ANALYSIS OF STABLE WEIGHT DECAY

In this section, we conducted comprehensive experiments to verify the advantage of the SWD scheduler over Decoupled Weight Decay and $L_2$ for Adam. Our experiments also include popular Adam variants, including AMSGrad (Reddi et al., 2019), Yogi (Zaheer et al., 2018), AdaBound (Luo et al., 2019), Padam (Chen and Gu, 2018), and RAdam (Liu et al., 2019),.

**Models and Datasets.** We choose Adam as the base optimizer, and train popular deep models, including ResNet18/34/50 (He et al., 2016), VGG16 (Simonyan and Zisserman, 2014), DenseNet121 (Huang et al., 2017), GoogLeNet (Szegedy et al., 2015), and Long Short-Term Memory (LSTM) (Hochreiter and Schmidhuber, 1997), on CIFAR-10/CIFAR-100 (Krizhevsky and Hinton, 2009), and Penn TreeBank (Marcus et al., 1993). The implementation details are in Appendix B.

**Image Classification on CIFAR-10/CIFAR-100.** Figure 4 shows the learning curves of AdamS, AdamW, and Adam on several benchmarks. In our experiments, AdamS always leads to lower test errors. Figure 5 shows that, even if with similar or higher training losses, AdamS still generalizes significantly better than AdamW, Adam, and recent Adam variants. Figure 6 displays the learning curves of all adaptive gradient methods. The test performance of other models can be found in Table 1. Simply scheduling weight decay for Adam by SWD even outperforms complex Adam variants. We also note that most Adam variants surprisingly generalize worse than SGD (See Appendix C).

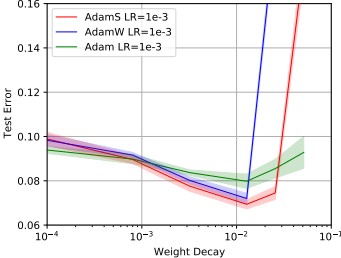 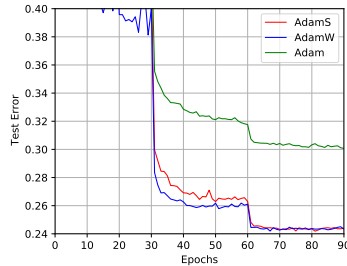

Figure 9: Rule 2 holds well for all Adam, AdamW, and AdamS on ResNet18. Figure 16 shows similar results for VGG16, which has no scale-invariant loss landscape.

Figure 10: ResNet50 on ImageNet. The lowest Top-1 test errors of AdamS, AdamW, and Adam are 24.19%, 24.29%, and 30.07%, respectively.

**Robustness to the hyperparameters.** Figure 7 further demonstrates that AdamS consistently outperforms Adam and AdamW under various weight decay hyperparameters. According to Figure 7, we also notice that the optimal decoupled weight decay hyperparameter in AdamW can be very different from $L_2$ regularization and SWD. Thus, AdamW requires re-tuning the weight decay hyperparameter in practice, which is time-consuming. Figure 8 shows that AdamS is more robust to the learning rate and weight decay than AdamW and Adam. AdamS has a much deeper and wider basin than AdamW and Adam. We leave the experiments with cosine annealing schedulers and warm restarts (Loshchilov and Hutter, 2016) in Appendix E, which also support that AdamS yields superior test performance and converges to lower training losses.

**Large-Batch Training.** In Figure 9, we again verified that Rule 2 holds well on all of Adam, AdamW, and AdamS. We also note that, in the case that applies Rule 2, AdamS generalizes better than AdamW and Adam on large-batch training ($B = 16384$).

**Limitations.** Figure 10 shows that, on ImageNet, the improvement of AdamS over AdamW becomes marginal. Figure 11 in Appendix C demonstrates that, for LSTM on Penn TreeBank, $L_2$ regularization yields better test results than both Decoupled Weight Decay and SWD.

## 6 CONCLUSION AND DISCUSSION

Our experiments on large-batch training shows that the learning-rate linear scaling rule sometimes is harmful to large-batch training due to bad convergence. We propose the weight-decay linear scaling rule for large-batch training as an alternative solution. We note that the results may not be explained by previous papers(Arora et al., 2019; Li et al., 2020), which studied scale-invariant deep loss landscape due to batch normalization(Ioffe and Szegedy, 2015). Instead, the empirical results verified our theoretical analysis with or without scale-invariant loss functions.

Inspired by the theoretical analysis of weight decay on stability and convergence, we propose a scheduler for weight decay. Previous work did not studied how to design schedulers for weight decay. Our empirical results demonstrate that SWD often makes improvements over $L_2$ regularization and Decoupled Weight Decay. The generalization gap between SGD and Adam can be almost closed by a simple weight decay scheduler on CIFAR datasets. Although our analysis mainly focused on Adam, SWD can be easily combined with other Adam variants, too.

Finally, we discuss the limitations of SWD. While SWD makes significant improvements in first three experiments of Section 5, the experiments on ImageNet and Language Modeling suggest that the improvement of SWD may be marginal for complex loss landscapes. SWD as the first practical design for weight decay scheduling is not perfect. In future, it will be interesting to design weight decay schedulers that can accelerate convergence in the early training phase while learning stable and accurate minima in the late training phase.

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

## A  PROOFS

### A.1  PROOF OF THEOREM 1

*Proof.* We first write the regularized loss function corresponding to SGD with vanilla weight decay as

$$f_t(\theta) = L(\theta) + \frac{\lambda'}{2\eta_t}\|\theta\|^2 \tag{12}$$

at $t$-th step.

If the corresponding $L_2$ regularization $\frac{\lambda'}{2\eta_t}$ is unstable during training, the regularized loss function $f_t(\theta)$ will also be a time-dependent function and has no non-zero stable stationary points.

Suppose we have a non-zero solution $\theta^\star$ which is a stationary point of $f(\theta, t)$ at $t$-th step and SGD finds $\theta_t = \theta^\star$ at $t$-th step.

Even if the gradient of $f_t(\theta)$ at $t$-step is zero, we have the gradient at $(t+1)$-th step as

$$g_{t+1} = \nabla f_{t+1}(\theta^\star) = \lambda'(\eta_t^{-1} - \eta_{t+1}^{-1})\theta^\star. \tag{13}$$

It means that $\|g_{t+1}\|^2 = \lambda'^2(\eta_t^{-1} - \eta_{t+1}^{-1})^2\|\theta^\star\|^2$.

To achieve convergence, we must have $\|g_{t+1}\|^2 = 0$.

It requires $(\eta_t^{-1} - \eta_{t+1}^{-1})^2 = 0$ or $\|\theta^\star\|^2 = 0$.

Theorem 2.2 of Shapiro and Wardi (1996) told us that the learning rate should be small enough for convergence. Obviously, we have $\eta < \infty$ in practice.

As $\eta_t = \eta_{t+1}$ does not hold, SGD cannot converging to any non-zero stationary point.

The proof is now complete. □

### A.2  PROOF OF THEOREM 2

*Proof.* Under Assumption 1, we write the dynamics of SGD with weight decay as

$$\theta_t = \theta_{t-1} - \eta_t H(\theta_{t-1} - \theta^\star) - \eta_t \xi_t - \eta_t \lambda \theta_{t-1} \tag{14}$$

Note that the stochastic gradient noise $\xi$ is zero-mean, $\mathbb{E}[\xi] = 0$, where $\xi$ is the difference of the true gradient and the stochastic gradient. The zero-mean property generally holds because SGD randomly selects minibatches from the whole training dataset, which is widely used in related papers (Xie et al., 2021c; Kingma and Ba, 2015; Reddi et al., 2019).

The dynamics above is equivalent to minimizing the training loss with $L_2$ regularization as

$$\theta_t = \theta_{t-1} - \eta_t \nabla L_{wd}(\theta) - \eta_t \xi_t, \tag{15}$$

where

$$\begin{aligned} L_{wd}(\theta) &= \frac{1}{2}(\theta - \theta^\star)^\top H(\theta - \theta^\star) + \lambda\theta^\top\theta \\ &= \frac{1}{2}(H + \lambda I)[\theta - H(H + \lambda I)^{-1}\theta^\star]^2 + H\lambda(H + \lambda I)^{-1}\theta^\star. \end{aligned} \tag{16}$$

Thus, we may rewrite the dynamics of SGD with weight decay as

$$\theta_t = \theta_{t-1} - \eta_t(H + \lambda I)[\theta_{t-1} - H(H + \lambda I)^{-1}\theta^\star] - \eta_t\xi_t \quad (17)$$

$$\theta_t - H(H + \lambda I)^{-1}\theta^\star = [I - \eta_t(H + \lambda I)](\theta_{t-1} - H(H + \lambda I)^{-1}\theta^\star) - \eta_t\xi_t \quad (18)$$

$$\mathbb{E}[\theta_t - H(H + \lambda I)^{-1}\theta^\star] = \mathbb{E}\left[\prod_{k=1}^{t}[I - \eta_k(H + \lambda I)](\theta_0 - H(H + \lambda I)^{-1}\theta^\star)\right], \quad (19)$$

where the expectation of the stochastic gradient noise term is zero. As we care about the difference between the expected learned solution and the true minimum $\theta^\star$, we may further obtain

$$\mathbb{E}[\theta_t - H(H + \lambda I)^{-1}\theta^\star] = \prod_{k=1}^{t}[I - \eta_k(H + \lambda I)](\theta_0 - H(H + \lambda I)^{-1}\theta^\star) - \lambda(H + \lambda I)^{-1}. \tag{20}$$

In the case of the constant learning rate scheduler, we may simplify the equation above as

$$\mathbb{E}[\theta_t - H(H + \lambda I)^{-1}\theta^\star] = [I - \eta(H + \lambda I)]^t(\theta_0 - H(H + \lambda I)^{-1}\theta^\star) - \lambda(H + \lambda I)^{-1}. \quad (21)$$

The proof is now complete. $\qquad\square$

## B  EXPERIMENTAL DETAILS

**Computational environment.** The experiments are conducted on a computing cluster with GPUs of NVIDIA® Tesla™ P100 16GB and CPUs of Intel® Xeon® CPU E5-2640 v3 @ 2.60GHz.

### B.1  IMAGE CLASSIFICATION ON CIFAR-10 AND CIFAR-100

**Data Preprocessing For CIFAR-10 and CIFAR-100:** We perform the common per-pixel zero-mean unit-variance normalization, horizontal random flip, and $32 \times 32$ random crops after padding with 4 pixels on each side.

**Hyperparameter Settings:** We select the optimal learning rate for each experiment from $\{0.0001, 0.001, 0.01, 0.1, 1, 10\}$ for non-adaptive gradient methods. We use the default learning rate for adaptive gradient methods in the experiments of Table **??**, while we compared Adam, AdamW, AdanS under various learning rates and batch sizes in the experiments of Figure 7 and Figure 8. In the experiments on CIFAF-10 and CIFAR-100: $\eta = 0.1$ for SGD and SGDS; $\eta = 0.001$ for Adam, AdamW, AdamW, AMSGrad, AdamW, AdaBound, and RAdam; $\eta = 0.01$ for Padam. For the learning rate schedule, the learning rate is divided by 10 at the epoch of $\{80, 160\}$ for CIFAR-10 and $\{100, 150\}$ for CIFAR-100, respectively. The batch size is set to 128 for both CIFAR-10 and CIFAR-100, unless we specify it on large-batch training.

The strength of $L_2$ regularization and SWD is default to 0.0005 as the baseline. Considering the linear scaling rule, we choose $\lambda_W = \frac{\lambda_{L_2}}{\eta}$. Thus, the weight decay of AdamW uses $\lambda_W = 0.5$ for CIFAR-10 and CIFAR-100. The basic principle of choosing weight decay strength is to let all optimizers have similar convergence speed.

We set the momentum hyperparameter $\beta_1 = 0.9$ for SGD and SGDS. As for other optimizer hyperparameters, we apply the default hyperparameter settings directly.

We repeated each experiment for three times in the presence of the error bars.

We leave the empirical results with the weight decay setting $\lambda = 0.0001$ in Appendix C.

### B.2  LARGE-BATCH TRAINING ON CIFAR-10

**Model:** we always used the common ResNet18 for large-batch training.

**Optimizer:** we always used Adam and its variant as the optimizer, because adaptive gradient methods often outperform SGD in large-batch training.

**The learning rate scheduler:** Suppose the number of epochs is $E$, the learning rate is divided by 10 at the epoch of $\{0.4E, 0.8E\}$.

The default batch size is 16384 in large-batch training. The default weight decay is 0.0001 or follows Rule 2. The default learning rate is 0.001 or follows Rule 1.

As for other optimizer hyperparameters, we apply the default hyperparameter setting directly.

### B.3 IMAGE CLASSIFICATION ON IMAGENET

**Data Preprocessing For ImageNet:** For ImageNet, we perform the per-pixel zero-mean unit-variance normalization, horizontal random flip, and the resized random crops where the random size (of 0.08 to 1.0) of the original size and a random aspect ratio (of $\frac{3}{4}$ to $\frac{4}{3}$) of the original aspect ratio is made.

**Hyperparameter Settings for ImageNet:** We select the optimal learning rate for each experiment from $\{0.0001, 0.001, 0.01, 0.1, 1, 10\}$ for all tested optimizers. For the learning rate schedule, the learning rate is divided by 10 at the epoch of $\{30, 60\}$. We train each model for 90 epochs. The batch size is set to 256. The weight decay hyperparameter of AdamS, AdamW, Adam are chosen from $\{5 \times 10^{-6}, 5 \times 10^{-5}, 5 \times 10^{-4}, 5 \times 10^{-3}, 5 \times 10^{-2}\}$. As for other optimizer hyperparameters, we still apply the default hyperparameter settings directly.

### B.4 LANGUAGE MODELING

We use a classical language model, Long Short-Term Memory (LSTM) (Hochreiter and Schmidhuber, 1997) with 2 layers, 512 embedding dimensions, and 512 hidden dimensions, which has 14 million model parameters and is similar to the "medium LSTM" in Zaremba et al. (2014). Note that our baseline performance is better than the reported baseline performance in Zaremba et al. (2014). The benchmark task is the word-level Penn TreeBank (Marcus et al., 1993). We empirically compared AdamS, AdamW, and Adam under the common and same conditions.

**Hyperparameter Settings.** Batch Size: $B = 20$. BPTT Size: $bptt = 35$. Learning Rate: $\eta = 0.001$. We run the experiments under various weight decay selected from $\{10^{-4}, 5 \times 10^{-5}, 10^{-5}, 5 \times 10^{-6}, 10^{-6}, 5 \times 10^{-7}, 10^{-7}\}$. The dropout probability is set to 0.5. We clipped gradient norm to 1.

## C SUPPLEMENTARY FIGURES AND RESULTS OF ADAPTIVE GRADIENT METHODS

**Popular Adam variants often generalize worse than SGD.** A few Adam variants tried to fix the hidden problems in adaptive gradient methods, including AdamW Loshchilov and Hutter (2018), AMSGrad (Reddi et al., 2019) and Yogi (Zaheer et al., 2018). A recent line of research, such as AdaBound (Luo et al., 2019), Padam (Chen and Gu, 2018), and RAdam (Liu et al., 2019), believes controlling the adaptivity of learning rates may improve generalization. This line of research usually introduces extra hyperparameters to control the adaptivity, which requires more efforts in tuning hyperparameters. However, we and Zhang et al. (2019c) found that this argument is contradicted with our comparative experimental results (see Table 1). In our empirical analysis, most advanced Adam variants may narrow but not completely close the generalization gap between adaptive gradient methods and SGD. SGD with a fair weight decay hyperparameter as the baseline performance usually generalizes better than recent adaptive gradient methods. The main problem may lie in weight decay. SGD with weight decay $\lambda = 0.0001$, a common setting in related papers, is often not a good baseline, as $\lambda = 0.0005$ often shows better generalization on CIFAR-10 and CIFAR-100 (see in Figures 19 and 21 of Appendix F). We also conduct comparative experiments with $\lambda = 0.0001$ (see Table 2 of Appendix C). Under the setting $\lambda = 0.0001$, while some existing Adam variants may outperform SGD sometimes due to the lower baseline performance of SGD, AdamS shows superior test performance. For example, for ResNet18 on CIFAR-10, the test error of AdamS is lower than SGD by nearly one point and no other Adam variant may compare with AdamS.

**Language Modeling.** It is well-known that, different from computer vision tasks, the standard Adam (with $L_2$ regularization) is the most popular optimizer for language models. Figure 11 in Appendix C demonstrates that the conventional belief is true that the standard $L_2$ regularization yields better

Table 2: Test performance comparison of optimizers with $\lambda_{L_2} = \lambda_S = 0.0001$ and $\lambda_W = 0.1$, which is a common weight decay setting in related papers. AdamS still show better test performance than popular adaptive gradient methods and SGD.

| DATASET | MODEL | SGD | ADAMS | ADAM | AMSGRAD | ADAMW | ADABOUND | PADAM | YOGI | RADAM |
|---------|-------|-----|-------|------|---------|-------|----------|-------|------|-------|
| CIFAR-10 | RESNET18 | 5.58 | **4.69** | 6.08 | 5.72 | 5.33 | 6.87 | 5.83 | 5.43 | 5.81 |
| | VGG16 | 6.92 | **6.16** | 7.04 | 6.68 | 6.45 | 7.33 | 6.74 | 6.69 | 6.73 |
| CIFAR-100 | RESNET34 | 24.92 | **23.50** | 25.56 | 24.74 | 23.61 | 25.67 | 25.39 | 23.72 | 25.65 |
| | DENSENET121 | **20.98** | 21.35 | 24.39 | 22.80 | 22.23 | 24.23 | 22.26 | 22.40 | 22.40 |
| | GOOGLENET | 21.89 | **21.60** | 24.60 | 24.05 | 21.71 | 25.03 | 26.69 | 22.56 | 22.35 |

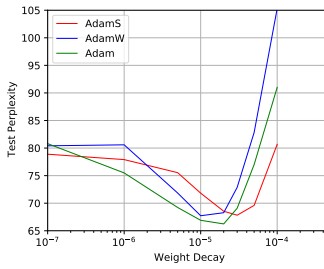

Figure 11: Language modeling under various weight decay. Note that the lower perplexity is better.

test results than both Decoupled Weight Decay and SWD. The weight decay scheduler suitable for language models is an open problem.

We report the learning curves of all adaptive gradient methods in Figure 12. They shows that vanilla Adam with SWD can outperform other complex variants of Adam.

Figure 13 displays the scatter plot of training losses and test errors during final 40 epochs of training DenseNet121 on CIFAR-100.

Figure 14 displays the test performance of AdamS, AdamW, and Adam under various weight decay hyperparameters of ResNet34 on CIFAR-100.

We train ResNet18 on CIFAR-10 for 900 epochs to explore the performance limit of AdamS, AdamW, and Adam in Figure 15.

Note that ResNets have scale-invariant loss landscape, while VGGs have no scale-invariant loss landscape. Our theoretical analysis generally holds under Assumption 1, which is independent of the scale-invariant property of the loss landscape. Figure 16 further shows that, with or without normalization layers, our theoretical result can be empirically observed. This is beyond the intrinsic learning rate proposed by Li et al. (2020) which depends on the scale-invariant property of the loss function.

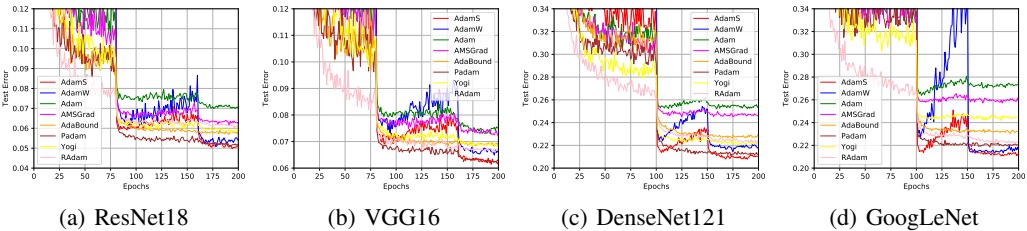

| (a) ResNet18 | (b) VGG16 | (c) DenseNet121 | (d) GoogLeNet |
|---|---|---|---|

Figure 12: The learning curves of adaptive gradient methods.

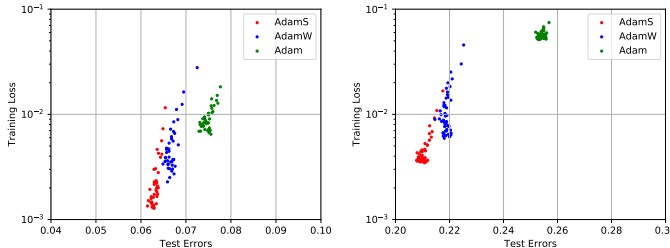

Figure 13: Even if with similar or higher training losses, AdamS still generalizes better than AdamW and Adam. The scatter plot of training losses and test errors during final 50 epochs of training VGG16 on CIFAR-10 and DenseNet121 on CIFAR-100.

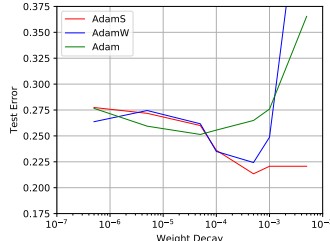

Figure 14: We compare the generalization of Adam, AdamW, and AdamS with various weight decay rates by training ResNet34 on CIFAR-100. The displayed weight decay of AdamW in the figure has been rescaled by the factor $\approx 0.001$. The optimal test performance of AdamS is significantly better than AdamW and Adam.

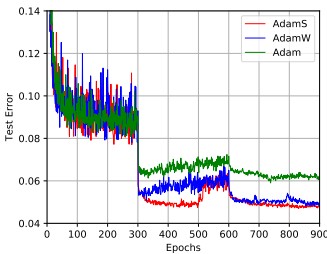

Figure 15: We train ResNet18 on CIFAR-10 for 900 epochs to explore the performance limit of AdamS, AdamW, and Adam. The learning rate is divided by 10 at the epoch of 300 and 600. AdamS achieves the most optimal test error, $4.70\%$.

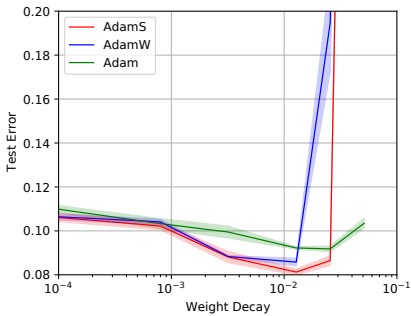

Figure 16: Rule 2 holds well for all of Adam, AdamW, and AdamS. VGG16 on CIFAR-10.

Table 3: In the experiment of ResNet18 trained via SGD on CIFAR-10, we verified that the optimal weight decay is approximately inverse to the number of epochs. The predicted optimal weight decay is approximately $0.1 \times \text{Epochs}^{-1}$, because the optimal weight decay is $\lambda = 0.0005$ selected from $\{10^{-2}, 5 \times 10^{-3}, 10^{-3}, 5 \times 10^{-4}, 10^{-4}, 5 \times 10^{-5}, 10^{-5}, 5 \times 10^{-6}, 10^{-6}\}$ with 200 epochs as the base case. The observed optimal weight decay is selected from $\{\text{Epochs}^{-1}, 0.1 \times \text{Epochs}^{-1}, 0.01 \times \text{Epochs}^{-1}\}$. We observed that the optimal test errors are all corresponding to the predicted optimal weight decay $\lambda = 0.1 \times \text{Epochs}^{-1}$. At least in the sense of the order of magnitude, the predicted optimal weight decay is fully consistent with the observed optimal weight decay. Thus, the empirical results supports that the optimal weight decay is approximately inverse to the number of epochs in the common range of the number of epochs.

| EPOCHS | $\lambda = \text{Epochs}^{-1}$ | $\lambda = 0.1 \times \text{Epochs}^{-1}$ | $\lambda = 0.01 \times \text{Epochs}^{-1}$ |
|---|---|---|---|
| 50 | 74.06 | **7.12** | 7.50 |
| 100 | 22.04 | **5.56** | 6.01 |
| 200 | 11.81 | **5.02** | 5.61 |
| 1000 | 4.67 | **4.43** | 6.02 |
| 2000 | 4.59 | **4.48** | 5.70 |

# D  ADDITIONAL ALGORITHMS

Algorithm 3 is the TensorFlow implementation for SGD.

Algorithm 4 is the implementation of Adai, AdaiW, and AdaiS. As $\frac{\beta_3}{1-\beta_1}$ is always 1 in Adai, AdaiW is identical to AdaiS.

We note that the implementation of AMSGrad in Algorithm 5 is the popular implementation in PyTorch. We use the PyTorch implementation in our paper, as it is widely used in practice.

---

**Algorithm 3: SGD in TensorFlow**

$g_t = \nabla L(\theta_{t-1}) + \lambda \theta_{t-1}$;
$m_t = \beta_1 m_{t-1} - \eta g_t$;
$\theta_t = \theta_{t-1} + m_t$;

---

**Algorithm 4: Adai /AdaiS=AdaiW**

$g_t = \nabla L(\theta_{t-1}) + \lambda \theta_{t-1}$;
$v_t = \beta_2 v_{t-1} + (1 - \beta_2)g_t^2$;
$\hat{v}_t = \frac{v_t}{1-\beta_2^t}$;
$\bar{v}_t = mean(\hat{v}_t)$;
$\beta_{1t} = (1 - \beta_0 \frac{\hat{v}_t}{\bar{v}_t}).Clip(0, 1 - \epsilon)$;
$m_t = \beta_{1t} m_{t-1} + (1 - \beta_{1t})g_t$;
$\hat{m}_t = \frac{m_t}{1-\prod_{k=1}^{t}\beta_{1k}}$;
$\theta_t = \theta_{t-1} - \eta\hat{m}_t - \eta\lambda\theta_{t-1}$;

---

**Algorithm 5: AMSGrad/AMSGradW**

$g_t = \nabla L(\theta_{t-1}) + \lambda \theta_{t-1}$;
$m_t = \beta_1 m_{t-1} + (1 - \beta_1)g_t$;
$v_t = \beta_2 v_{t-1} + (1 - \beta_2)g_t^2$;
$\hat{m}_t = \frac{m_t}{1-\beta_1^t}$;
$v_{max} = max(v_t, v_{max})$;
$\hat{v}_t = \frac{v_{max}}{1-\beta_2^t}$;
$\theta_t = \theta_{t-1} - \frac{\eta}{\sqrt{\hat{v}_t}+\epsilon}\hat{m}_t - \eta\lambda\theta_{t-1}$;

---

**Algorithm 6: AMSGradS**

$g_t = \nabla L(\theta_{t-1})$;
$m_t = \beta_1 m_{t-1} + (1 - \beta_1)g_t$;
$v_t = \beta_2 v_{t-1} + (1 - \beta_2)g_t^2$;
$\hat{m}_t = \frac{m_t}{1-\beta_1^t}$;
$v_{max} = max(v_t, v_{max})$;
$\hat{v}_t = \frac{v_{max}}{1-\beta_2^t}$;
$\bar{v}_t = mean(\hat{v}_t)$;
$\theta_t = \theta_{t-1} - \frac{\eta}{\sqrt{\hat{v}_t}+\epsilon}\hat{m}_t - \frac{\eta}{\sqrt{\bar{v}_t}}\lambda\theta_{t-1}$;

---

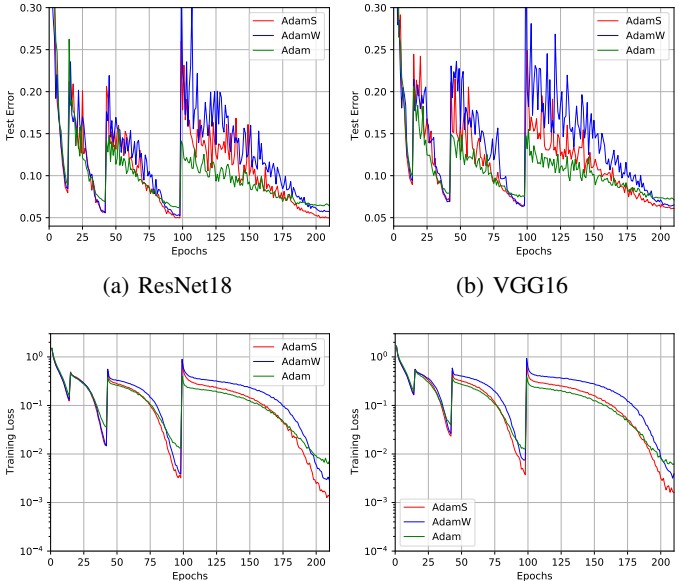

Figure 17: The learning curves of ResNet18 and VGG16 on CIFAR-10 with cosine annealing and warm restart schedulers. The weight decay hyperparameter: $\lambda_{L_2} = \lambda_S = 0.0005$ and $\lambda_W = 0.5$. Top Row: Test curves. Bottom Row: Training curves. AdamS yields significantly lower test errors and training losses than AdamW and Adam.

## E   SUPPLEMENTARY EXPERIMENTS WITH COSINE ANNEALING SCHEDULERS AND WARM RESTARTS

In this section, we conducted comparative experiments on AdamS, AdamW, and Adam in the presence of cosine annealing schedulers and warm restarts proposed by Loshchilov and Hutter (2016). We set the learning rate scheduler with a recommended setting of Loshchilov and Hutter (2016): $T_0 = 14$ and $T_{mul} = 2$. The number of total epochs is 210. Thus, we trained each deep network for four runs of warm restarts, where the four runs have 14, 28, 56, and 112 epochs, respectively. Other hyperparameters and details are displayed in Appendix B.

Our experimental results in Figures 17 and 18 suggest that AdamS consistently outperforms AdamW and Adam in the presence of cosine annealing schedulers and warm restarts. It demonstrates that, with various learning rate schedulers, the advantage of SWD may generally hold.

Moreover, we did not empirically observe that cosine annealing schedulers with warm restarts may consistently outperform the common piecewise-constant learning rate schedulers for adaptive gradient methods. We noticed that Loshchilov and Hutter (2016) empirically compared four-staged piecewise-constant learning rate schedulers with cosine annealing schedulers with warm restarts, and argue that cosine annealing schedulers with warm restarts are better. There may be two possible causes. First, three-staged piecewise-constant learning rate schedulers, which usually have a longer first stage and decay learning rates by multiplying 0.1, are the recommended settings, while the four-staged piecewise-constant learning rate schedulers in Loshchilov and Hutter (2016) are usually not optimal. Second, warm restarts may be helpful, while cosine annealing may be not. The ablation study on piecewise-constant learning rate schedulers with warm restarts is lacked. We argued that how to choose learning rate schedulers may still be an open question, considering the complex choices of schedulers and the complex loss landscapes.

## F   WEIGHT DECAY IN MOMENTUM

In this section, we rethink weight decay in Momentum.

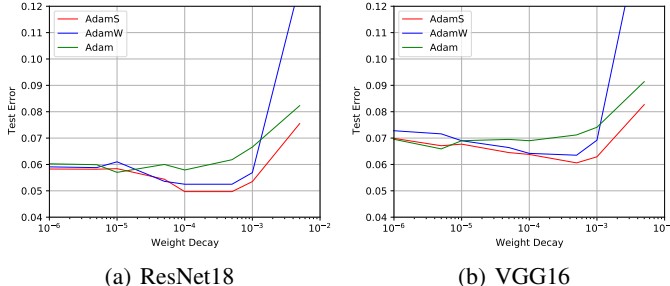

(a) ResNet18           (b) VGG16

Figure 18: The test errors of ResNet18 and VGG16 on CIFAR-10 under various weight decay with cosine annealing and warm restart schedulers. AdamS yields significantly better optimal test performance than AdamW and Adam.

It is also commonly believed that $L_2$ regularization is an optimal weight decay implementation when Adaptive Learning Rate is not involved. Almost all deep learning libraries directly use $L_2$ regularization as the default weight decay implementation. However, we reveal that $L_2$ regularization is not identical to decoupled weight decay Loshchilov and Hutter (2018) and sometimes harms performance slightly when Momentum is involved.

We take Stochastic Heavy Ball (SHB) (Zavriev and Kostyuk, 1993), which only uses fixed momentum inertia and dampening coefficients, as the studied example in the presence of Momentum, as SHB-style Momentum methods are widely used by many popular optimizers. SGD implemented in PyTorch (Paszke et al., 2019) is actually SHB with default hyperparameters. We write SHB with $L_2$ regularization and SHB with decoupled weight decay in Algorithm 7.

| **Algorithm 7:** SGD/SGDW (SHB/SHBW) | **Algorithm 8:** SGDS (SHBS) |
|---|---|
| $g_t = \nabla L(\theta_{t-1}) + \lambda\theta_{t-1};$ 
 $m_t = \beta_1 m_{t-1} + \beta_3 g_t;$ 
 $\theta_t = \theta_{t-1} - \eta m_t - \eta\lambda\theta_{t-1};$ | $g_t = \nabla L(\theta_{t-1});$ 
 $m_t = \beta_1 m_{t-1} + \beta_3 g_t;$ 
 $\theta_t = \theta_{t-1} - \eta m_t - \frac{\beta_3}{1-\beta_1}\eta\lambda\theta_{t-1};$ |

Inspired by Theorem 1, we propose SGD with SWD (SGDS) as

$$\begin{cases} m_t = \beta_1 m_{t-1} + \beta_3 g_t \\ \theta_t = \left[1 - \frac{\beta_3(1-\beta_1^t)}{1-\beta_1}\eta\lambda\right]\theta_{t-1} - \eta m_t, \end{cases} \tag{22}$$

where $\frac{\beta_3(1-\beta_1^t)}{1-\beta_1}\eta$ is the effective learning rate in SHB, as $m_t \approx \frac{\beta_3(1-\beta_1^t)}{1-\beta_1}\mathbb{E}[g_t]$. As $(1-\beta_1^t)$ converges to 1 soon, we use the simplified $\frac{\beta_3}{1-\beta_1}\eta$ as the bias correction in practice, where relatively large weight decay in the first dozen iterations can work like a model parameter initialization strategy. It is easy to see that the factor $\frac{\beta_3}{1-\beta_1}$ for weight decay is exactly the difference between our stable weight decay and decoupled weight decay suggested by Loshchilov and Hutter (2018). The pseudocode of SGDS is displayed in Algorithm 8. It may also be called SHB with SWD (SHBS).

**Empirical Analysis on SGDS.** We empirically verified that the optimal performance of SGDS/SGDW is often slightly better than the widely used vanilla SGD and SGD, as seen in Figures 19 and 21. We leave more empirical results of SGDS in Appendix G (see Table 4).

In Figure 19, the optimal performance of SGDS/SGDW is better than the widely used Vanilla SGD and SGD. For Vanilla SGD, SGD, and SGDS, we may choose $\lambda_{L_2} = \lambda_S = 0.0005$ to maintain the optimal performance. But we have to re-tune $\lambda_W = 0.005$ for SGDW. Hyperparameter Setting: $\beta_1 = 0$ for Vanilla SGD; $\beta_1 = 0.9$ for SGD, SGDW, and SGDS. We repeat each simulation for three runs. A similar experimental result for ResNet18 is presented in Appendix G. In Figure 20, we report the test performance of SGDS and SGD for ResNet18 on CIFAR-10 under various learning rates and weight decay. SGDS has a deeper blue basin than SGD.

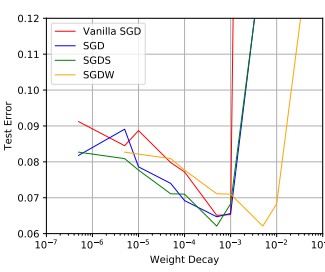 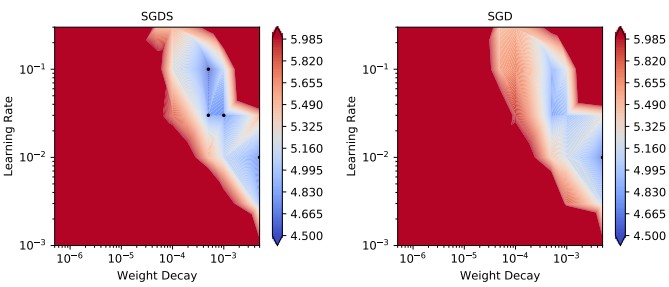

Figure 19: We compare the generalization of Vanilla SGD, SGD, SGDW, and SGDS under various weight decay hyperparameters by training VGG16 on CIFAR-10.

Figure 20: The test errors of ResNet18 on CIFAR-10. SGDS has a slightly deeper blue basin near dark points ($\leq 4.83\%$). The optimal choices of $\eta$ and $\lambda$ are very close for SGDS and SGD.

Table 4: Test performance comparison of Adai, AdaiS, SGD, and SGDS. Stable/Decoupled Weight Decay often outperform $L_2$ regularization for optimizers involving in momentum. We report the mean and the standard deviations (as the subscripts) of the optimal test errors computed over three runs of each experiment.

| DATASET | MODEL | ADAIS | ADAI | SGDS | SGD |
|---------|-------|-------|------|------|-----|
| CIFAR-10 | RESNET18 | $\mathbf{4.59}_{0.16}$ | $4.74_{0.14}$ | $4.69_{0.09}$ | $5.01_{0.03}$ |
| | VGG16 | $\mathbf{5.81}_{0.07}$ | $6.00_{0.09}$ | $6.28_{0.07}$ | $6.42_{0.02}$ |
| CIFAR-100 | DENSENET121 | $\mathbf{19.44}_{0.21}$ | $19.59_{0.38}$ | $19.61_{0.26}$ | $19.81_{0.33}$ |
| | GOOGLENET | $\mathbf{20.50}_{0.25}$ | $20.55_{0.32}$ | $20.68_{0.03}$ | $21.21_{0.29}$ |

It is not well-known that SGDW often outperforms SGD slightly. We believe it is mainly because people rarely know re-tuning $\lambda_W$ based on $\frac{\beta_3}{1-\beta_1}$ is necessary for maintaining good performance when people switch from SGD to SGDW. As the effective learning rate of SGD is $\frac{\beta_3}{1-\beta_1}\eta$, the weight decay rate of SGDW is actually $R = \frac{1-\beta_1}{\beta_3}\lambda_W$ rather than $\lambda_W$. If we use different settings of $\beta_1$ and $\beta_3$ in decoupled weight decay, we will undesirably change the weight decay rate $R$ unless we re-tune $\lambda_W$. However, people usually directly let $\lambda_W = \lambda_{L_2}$ for SGDW in practice. Figure 19 shows that, the optimal $\lambda_{L_2}$ and $\lambda_S$ in SGD ( with $\beta_1 = 0.9$ and $\beta_3 = 1$) are both 0.0005, while the optimal $\lambda_W$ is 0.005 instead. The optimal $\lambda_{L_2}$ and $\lambda_S$ are almost same, while the optimal $\lambda_W$ is quite different. Thus, the advantage of SGDS over SGDW can save us from re-tuning the weight decay hyperparameter.

## G SUPPLEMENTARY RESULTS OF SGD WITH MOMENTUM AND ADAI

We compare SGDS, SGDW, vanilla SGD and SGD under various weight decay hyperparameters by training ResNet18 on CIFAR-10. We observe that the optimal performance of SGDS/SGDW is better than vanilla SGD and SGD in Figure 21. The advantage of SGDS over SGDW is that we do not need to fine-tune the weight decay hyperparameters based on $\frac{\beta_3}{1-\beta_1}$.

We report the learning curves of SGD and SGDS in Figure 22. SGDS compares favorably with SGD.

Based on a diffusion theoretical framework (Xie et al., 2021b), Xie et al. (2020) proposed Adaptive Inertia Estimation (Adai) that uses adaptive momentum inertia instead of Adaptive Learning Rate to help training. Adaptive inertia can be regarded as an inertia-adaptive variant of SHB. The previous analysis on SHB can be easily generalized to Adai. We display Adai with SWD (AdaiS) in Algorithm

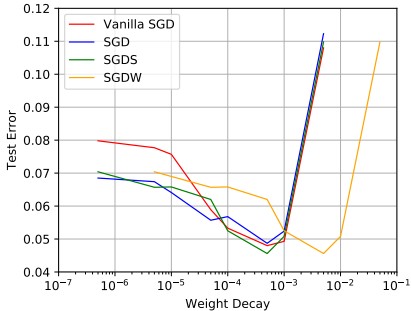

Figure 21: We compare the generalization of Vanilla SGD, SGD, SGDW, and SGDS with various weight decay hyperparameters by training ResNet18 on CIFAR-10. The optimal weight decay rates are near 0.0005 for all three weight implementations. The optimal performance of SGDS/SGDW is better than Vanilla SGD and SGD. For Vanilla SGD, SGD, and SGDS, we may safely choose $\lambda_{L_2} = \lambda_S = 0.0005$. But we have to re-tune $\lambda_W = 0.005$ for SGDW. Hyperparameter Setting: $\beta_1 = 0$ for Vanilla SGD; $\beta_1 = 0.9$ for SGD, SGDW, and SGDS.

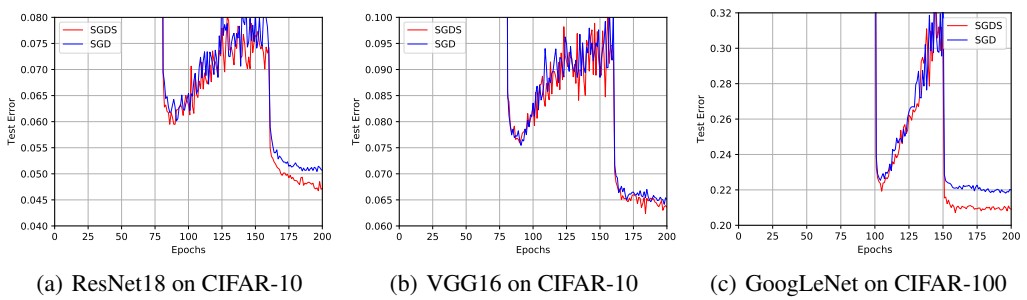

(a) ResNet18 on CIFAR-10     (b) VGG16 on CIFAR-10     (c) GoogLeNet on CIFAR-100

Figure 22: Generalization analysis on SGDS and SGD with $L_2$ regularization. Hyperparameter Setting: $\lambda_S = \lambda_{L_2} = 0.0005$ and $\beta_1 = 0.9$.

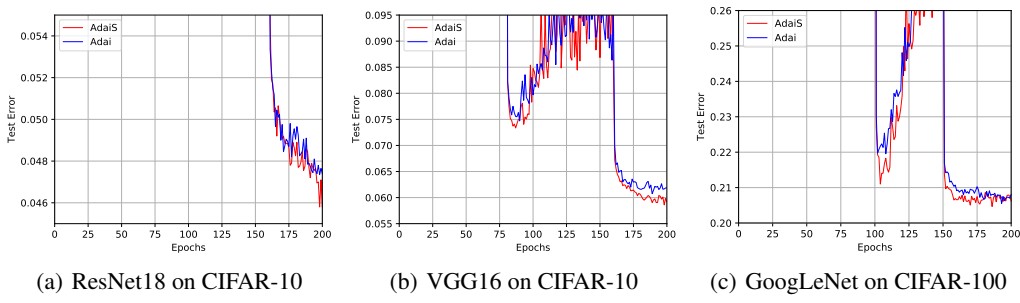

(a) ResNet18 on CIFAR-10     (b) VGG16 on CIFAR-10     (c) GoogLeNet on CIFAR-100

Figure 23: Generalization analysis on AdaiS and Adai with $L_2$ regularization. Hyperparameter Setting: $\lambda_S = \lambda_{L_2} = 0.0005$.

4. We report the learning curves of Adai and AdaiS in Figure 23, which may verify the generalization advantage of AdaiS over Adai.

