# OpenReview forum: "Understanding and Scheduling Weight Decay"
_ICLR.cc/2022/Conference — ICLR 2022 Submitted_

### Official Review · Reviewer_saHk · 2021-10-26

**Correctness:** 3
**Technical Novelty And Significance:** 2
**Empirical Novelty And Significance:** 2
**Recommendation:** 3
**Confidence:** 4

**Main Review:**

# Strength:

The organization of this paper is quite clear. The theoretical analysis in this paper is easy to follow. In the empirical study, the proposed AdamS is shown to be superior to Adam and AdamW in various datasets and neural networks. Overall, the quality of this paper is satisfactory in terms of clarity and empirical evaluation.

# Weakness:

My concern for this paper is that there are some gaps between the theoretical assumptions and experiment settings.

One major issue is about Assumption 1 and the scale-invariant property of some neural architectures used in the experiment. The loss function is a scale-variant function in Assumption 1 but ResNet and DenseNet are both scale-invariant networks. The scale property matters here since weight decay has its main effect in controlling the weight magnitude. Theorems in this paper only rely on the quadratic approximation in Assumption 1. But the practically tested networks violate Assumption 1.

In addition, there are many existing papers (Hoffer et al., 2018, Arora et al., 2019, Li et al., 2020, Li et al., 2020) that investigate the effect of weight decay in deep neural networks with batch norm. Some of the observed phenomena can be explained by existing theories that respect the scale invariant property of ResNets, see the comments below. This puts the the novelty of theoretical analyses in this paper in a disadvantageous position.

Here are some specific comments.

1. $\eta_t$=$\eta_2$ in step-wise learning rate decay, except for 2 or 3 steps, which should not be an issue for stability of stationary points. I notice that the experiment in this paper uses the stepwise learning rate decay. How does the stable $\eta$ in stepwise learning rate decay lead to unstable training?

2. The paper claim that *Fig. 2 suggests given long enough training time, the optimal weight decay hyperparameter tends to be zero.* However, I think Fig.2 does not show the *optimal* learning rate tends to be zero when the training time is long. To support the claim, the figure should show the result of the same long training time using different weight decay.

3. Figure 3 shows that increasing weight decay is more effective than increasing learning rate when a large batch size is used. This phenomenon can be explained by the effective learning rate argument in scale-invariant neural networks, which claims that increasing weight decay has the effect of increasing the learning rate. The experiment in Fig. 3 is done using a ResNet, i.e. a scale-invariant network. So I wonder if we have the same effect when a standard CNN like VGG is used.

4. The test error curves in this paper (Fig.4 and 5) show a severe overfitting effect when using AdamW, which is not consistent with my experience. Could the authors give some comments on this effect? Is there any existing paper that shows the same overfitting effect of AdamW?

5. Equation (20) and (21) seem to have typos about $\theta^*$ in both right and left hand sides.

In summary, I would say that I enjoy the paper's clarity and structure but some main technical issues prevent it being published as its current status.Thus, I suggest settling the conflicts in theory and practice in an updated version.


[1] Arora, Sanjeev, Kaifeng Lyu, and Zhiyuan Li. "Theoretical analysis of auto rate-tuning by batch normalization." International Conference on Learning Representations. 2019. https://arxiv.org/abs/1812.03981

[2] Li, Zhiyuan, and Sanjeev Arora. "An Exponential Learning Rate Schedule for Deep Learning." International Conference on Learning Representations. 2020. https://arxiv.org/abs/1910.07454

[3] Li, Zhiyuan, Kaifeng Lyu, and Sanjeev Arora. "Reconciling modern deep learning with traditional optimization analyses: The intrinsic learning rate." Neural Information Processing Systems. 2020. https://arxiv.org/pdf/2010.02916.pdf


**Summary Of The Paper:**

The paper proposes to understand the effect of weight decay and design schedules for weight decay in Adam based on the theory. The first contribution is to understand the dynamics of weight decay in a neural network by assuming the loss function is quadratic. The theoretical analysis shows that weight decay biases stationary points but accelerates convergence. Second, the paper proposes to increase the weight decay when a large batch size is used based on the convergence acceleration property of WD. Finally, AdamS is proposed to improve the stability of Adam and some empirical results show its effectiveness using various neural architectures on small-scale datasets (not good on ImageNet).

**Summary Of The Review:**

Despite the effectiveness of the proposed AdamS in small-scale datasets, the theoretical analysis about weight decay does not respect the scale invariant property of some important neural networks, which are used in the empirical validation of this paper to demonstrate some results of its analysis. This results in a severe conflict between theory and practice in this paper, so I would suggest a further revision for this manuscript.

---

> ### Author Response · Authors · 2021-11-22
> **Responses (1) to Reviewer saHk**
>
> We appreciate the reviewer for the hard work and constructive comments.
>
> In the revised manuscript, we will settle the conflicts in theory and practice the reviewer pointed out.
>
> The main concerns have been duly addressed below.
>
> Q1: About Assumption 1. The loss landscape of used networks, such as ResNets, violate Assumption 1.
> A1: Even if the loss landscape is scale-invariant, we may always apply the Taylor approximation to a deterministic loss function. We argue that, Assumption 1 (the second-order Taylor approximation) is very common in the papers on optimization dynamics of deep learning [1-5]. Our theoretical analysis under Assumption 1 can provides novel insights.
>
> Moreover, it is known that stochastic optimization can escape multiple bad minima [2,8-10] and finally converge to a good minimum during training. Thus, the second-order Taylor approximation can be repeatedly applied near each minimum during training. Theorem 2 proves a useful tool to understand convergence behaviors which may repeatedly happen during searching minima of deep learning
>
> Q2: Some of the observed phenomena can be explained by existing theories that respect the scale invariant property of ResNets, see the comments below. This puts the novelty of theoretical analyses in this paper in a disadvantageous position.
>
> A2: We argue that, our theoretical analysis is independent of the scale invariant property of the loss landscape. The theoretical analysis holds for neural networks with or without scale-invariant loss landscape. Moreover, the theoretical analysis inspired us to proposed Rule 2 and Stable Weight Decay, which are novel contributions beyond existing papers on scale-invariant loss.
>
> Q3: $\eta_{t} = \eta_{2}$ in step-wise learning rate decay, except for 2 or 3 steps, which should not be an issue for stability of stationary points. I notice that the experiment in this paper uses the stepwise learning rate decay. How does the stable $\eta$ in stepwise learning rate decay lead to unstable training?
>
> A3: The stable $\eta$ in stage-wise learning rate decay itself does not lead to unstable training.
> If we use vanilla weight decay as Figure 1 shows, the effective weight decay strength is $\frac{\lambda^{\prime}}{\eta_{t}}$, which may not be suitable for the whole training procedure. It means we are optimizing highly different training losses for different training stages.
> If we use AdamW and step-wise learning rate decay, the stability problem mainly comes from Adaptive Learning Rate.
>
> Q4: The paper claim that Fig. 2 suggests given long enough training time, the optimal weight decay hyperparameter tends to be zero. However, I think Fig.2 does not show the optimal weight decay tends to be zero when the training time is long. To support the claim, the figure should show the result of the same long training time using different weight decay.
>
> A4: We totally agree with your suggestion. While this statement is also supported by [6], we need to present more direct evidence for the statement. We will revised the statement to make it more precisely and present a new Table 3 to verify the original statement in the revised version.
>
> Q5: Figure 3 shows that increasing weight decay is more effective than increasing learning rate when a large batch size is used. This phenomenon can be explained by the effective learning rate argument in scale-invariant neural networks, which claims that increasing weight decay has the effect of increasing the learning rate. The experiment in Fig. 3 is done using a ResNet, i.e. a scale-invariant network. So I wonder if we have the same effect when a standard CNN like VGG is used.
>
> A5: Yes, we can observe a similar phenomenon on the experiments of VGG16. We will present the experimental results of VGG16 on large-batch training in Figure 16 and the last paragraph of Appendix C of the revised version. While it seems that the effective learning rate argument in scale-invariant neural networks can explain Rule 2 for scale-invariant neural networks, our experiments suggest that Rule 2 is independent of the scale-invariant property. Thus, our contribution is novel and beyond existing works.
>
> Please refer to Responses (2) for more discussions.

---

> ### Author Response · Authors · 2021-11-22
> **Responses (2) to Reviewer saHk**
>
> Q6: The test error curves in this paper (Fig.4 and 5) show a severe overfitting effect when using AdamW, which is not consistent with my experience. Could the authors give some comments on this effect? Is there any existing paper that shows the same overfitting effect of AdamW?
>
> A6: This is actually not real overfitting, because the test error still decreases during the final training stage. According to our experience, the phenomenon that test errors increases usually happens before the final learning rate decay. Figure 12 shows similar phenomenon for other optimizers. If we decrease the weight decay strength to a relatively small value, the phenomenon will be less significant or even disappear. The phenomenon seems like a result of regularization which is not bad. In the tasks where SGD outperforms Adam, SGD often has the same phenomenon, too [7].
>
> Q7: Equation (20) and (21) seem to have typos about $\theta^{\star}$ in both right and left hand sides.
>
> A7: We kindly note that they are not typos. Equations (20) and (21) describe how the expected difference between $\theta_{t}$ and $H (H + \lambda I )^{-1} \theta^{\star}$ evolves after $t$ iterations, where $\theta^{\star}$ is considered as the known minimum of the original loss function. Equations (20) and (21) have $\theta^{\star}$ in both right and left hand sides for capturing the expected difference more intuitively. Thanks for reading the details of our theoretical analysis.
>
> References:
>
> [1] Zhang, G., Li, L., Nado, Z., Martens, J., Sachdeva, S., Dahl, G., ... & Grosse, R. B. (2019). Which algorithmic choices matter at which batch sizes? insights from a noisy quadratic model. Advances in neural information processing systems, 32, 8196-8207.
>
> [2] Xie, Z., Sato, I., & Sugiyama, M. (2020, September). A Diffusion Theory For Deep Learning Dynamics: Stochastic Gradient Descent Exponentially Favors Flat Minima. In International Conference on Learning Representations.
>
> [3] Mandt, S., Hoffman, M. D., & Blei, D. M. (2017). Stochastic Gradient Descent as Approximate Bayesian Inference. Journal of Machine Learning Research, 18, 1-35.
>
> [4] Li, Q., Tai, C., & Weinan, E. (2017, July). Stochastic modified equations and adaptive stochastic gradient algorithms. In International Conference on Machine Learning (pp. 2101-2110). PMLR.
>
> [5] Zhou, P., Feng, J., Ma, C., Xiong, C., & Hoi, S. C. H. (2020). Towards Theoretically Understanding Why Sgd Generalizes Better Than Adam in Deep Learning. Advances in Neural Information Processing Systems, 33.
>
> [6] Lewkowycz, A., & Gur-Ari, G. (2020). On the training dynamics of deep networks with $ L_2 $ regularization. arXiv preprint arXiv:2006.08643. Advances in Neural Information Processing Systems, 33.
>
> [7] Chen, J., Zhou, D., Tang, Y., Yang, Z., Cao, Y., & Gu, Q. (2020, January). Closing the Generalization Gap of Adaptive Gradient Methods in Training Deep Neural Networks. In IJCAI.
>
> [8] Kleinberg, B., Li, Y., & Yuan, Y. (2018, July). An alternative view: When does SGD escape local minima?. In International Conference on Machine Learning (pp. 2698-2707). PMLR.
>
> [9] Zhu, Z., Wu, J., Yu, B., Wu, L., & Ma, J. (2019, May). The Anisotropic Noise in Stochastic Gradient Descent: Its Behavior of Escaping from Sharp Minima and Regularization Effects. In International Conference on Machine Learning (pp. 7654-7663). PMLR.
>
> [10] Jastrzębski, S., Kenton, Z., Arpit, D., Ballas, N., Fischer, A., Bengio, Y., & Storkey, A. (2017). Three factors influencing minima in sgd. arXiv preprint arXiv:1711.04623.

---

> > ### Comment · Reviewer_saHk · 2021-11-30
> > **Response to authors**
> >
> > Thanks for your response. I appreciate the efforts of the authors and some concerns are addressed in the response. Here are some following comments.
> >
> > About the novelty of theoretical analysis, the authors claim that the contribution is that the Rule 2 and Stable Weight Decay inspired by the analysis. For Rule 2, the experiment is done using Adam and shows that the optimal weight decay is around 1e-2. The demand for large weight decay is due to the second-order normalization in Adam from my point of view. So Rule 2 is not a significant contribution for me. For SWD, Table 1 does not show **significant** improvement of SWD as described by the paper (e.g. CIFAR100 result). In summary, my concern for the contribution remains after the rebuttal period.
> >
> > About Equation (20) and (21), if there is no typos, then I am much more confused by the proof. Equation (3) and (4) in Theorem 2 are results of $\theta_t$ converging to $\theta^*$. However, Equation (20) and (21), which are the proof for Theorem 2, show the convergence of $\theta_t$ to $H(H+\lambda I)^{-1}\theta^*$. Starting from Equation (19), it is straightforward to derive Equation (3) and (4). But the authors's response suggests that Equation (20) and (21) are the final result. Could the authors explain which result is important and why?
> >
> > Moreover, on the RHS of Equation (20) and (21), there is a vector minus a matrix ($\lambda(H+\lambda I)^{-1}$ without $\theta^*$). Could the authors show how to derive the $\lambda(H+\lambda I)^{-1}$ without $\theta^*$ and define the operation between vector and matrix to make the proof more clear?

---

> > > ### Author Response · Authors · 2021-11-30
> > > **Responses (3) to Reviewer saHk**
> > >
> > > We highly appreciate Reviewer saHk’s updated review and constructive comments.
> > >
> > > We would like to present three direct responses as follows.
> > >
> > > We believe that your main concerns have been addressed.
> > >
> > > We really hope Reviewer saHk may reconsider the responses and evaluate the contributions of our work.
> > >
> > >
> > > Q1: The demand for large weight decay is due to the second-order normalization in Adam from my point of view. So Rule 2 is not a significant contribution for me.
> > >
> > > A1: We kindly note that the demand for large weight decay is due to faster convergence rather than `` the second-order normalization in Adam’’. We presented the experimental results of Adam, because Adam usually outperforms SGD in large-batch training. Rule 2 holds for SGD, empirically and theoretically, too. Considering the importance and popularity of Rule 1, we believe Rule 2 is a novel and significant contribution to large-batch training.
> > >
> > > Q2: For SWD, Table 1 does not show significant improvement of SWD as described by the paper (e.g. CIFAR100 result).
> > >
> > > A2: We reported that AdamS outperforms AdamW by $0.83$ point in the CIFAR-100 results. The improvement is usually consider significant in comparisons of optimization methods. The improvement of SWD over decoupled weight decay is similarly significant to the advantage of SGD over Adam(W) on CIFAR-100.
> > >
> > > Q3: About the typos Equation (20) and (21).
> > >
> > > A3: We feel very sorry and apologize for misunderstanding your point in the previous response.
> > >
> > > There are typos in Equations (20) and (21). The left-hand side of Equations (20) and (21) should be the same as that of Equation (3) and (4).
> > >
> > > The right-hand size of Equations (20) and (21) should have the last term as $\lambda (H+\lambda I)^{-1}\theta^{\star}$. Thanks a lot for pointing out the typos.

---

### Official Review · Reviewer_1qbp · 2021-10-28

**Correctness:** 1
**Technical Novelty And Significance:** 2
**Empirical Novelty And Significance:** 3
**Recommendation:** 6
**Confidence:** 2

**Main Review:**

I think AdamS is interesting and might help close the gap between SGD and Adam, however given that Adam is not widely used for vision tasks, its applications are limited. In text, weight decay is not used as often, so it is not clear if there are applications there, it might be interesting to explore that further with a more realistic training set/architecture than TreeBank and LSTM.

AdamS improvements are not that large for ImageNet, but this baseline seems pretty strong since previously reported Adam behaviour is much lower (for example in https://arxiv.org/pdf/2002.11803.pdf)

- I like that AdamS is more robust to hyperparameters.

- the connection between section 2 and the rest seems a little weak.

-How does eq7 motivate rule 2? rule 1 and rule 2 have the same scaling with respect to eq7. Why does increasing weight decay improve convergence but not increasing the learning rate?

**Summary Of The Paper:**

The main contribution of the paper is to propose AdamS a new version of Adam with weight decay which has an implicit decay of the weight decay.

The contents of section 2 are quite standard: for GD linear regression with L2 regularization, its effect is mainly to shift the hessian: the minimum is not reached and the time to convergence depends on the eigenvalue of the (modified) hessian.

Sections 3,4,5 are roughly independent of 2 and they focus on improving Adam with weight decay. In section 3 they compare rescaling the learning rate vs weight decay with the batch size and they conclude that it is better to rescale the weight decay.  In section 4, they propose AdamS  which is more "stable" version of Adam.

**Summary Of The Review:**

It seems like AdamS is only beneficial to training and I think it might be helpful to add this modification whenever using Adam in combination with weight decay. The improvements are not superstrong but there does not seem to be any drawback.

---

> ### Author Response · Authors · 2021-11-22
> **Responses (1) to Reviewer 1qbp**
>
> We appreciate the reviewer for the helpful comments and the kind support to our work.
>
> The main concerns have been duly addressed below.
>
> Q1: I think AdamS is interesting and might help close the gap between SGD and Adam, however given that Adam is not widely used for vision tasks, its applications are limited. In text, weight decay is not used as often, so it is not clear if there are applications there, it might be interesting to explore that further with a more realistic training set/architecture than TreeBank and LSTM.
>
> A1: We kindly argue that AdamS has show advantages in the following three applications.
>
> First, we note that, in the experiments of large-batch training where Adam outperforms SGD, the proposed AdamS can still show the performance improvement over Adam with $L_{2}$ regularization and Adam with decoupled weight decay. For example, the test accuracy of Adam (90%) may significantly outperform SGD (87%) by three points for ResNet18 in our baseline large-batch experiments (B=16384). It is known that Adam/AdamW is the popular optimizer with convergence guarantees for large-batch training. Our experimental results in Figure 9 demonstrated AdamS outperforms Adam and AdamW on large-batch training of ResNet18. In the revised version, we also supplied the large-batch training experiments of VGG16, which also verified the advantage of SWD over $L_{2}$ regularization and decoupled weight decay.
>
> Second, as Figures 7 and 8 show, AdamS is more robust to the training hyperparameters than Adam and AdamW. This proves another practical advantage of SWD over $L_{2}$ regularization and decoupled weight decay for Adam, because tuning hyperparameters can be less time-consuming and difficult for AdamS.
>
> Third, Figure 17 further supports that AdamS converges better (to lower training losses) than both AdamW and Adam. A similar empirical evidence is used by [3] to prove the advantage of AdamW over Adam with $L_{2}$ regularization.
>
> We also agree that it will be interesting to further explore wide applications in language model along the proposed approach in future. Thanks for the suggestion.
>
> Q2: AdamS improvements are not that large for ImageNet, but this baseline seems pretty strong since previously reported Adam behaviour is much lower (for example in https://arxiv.org/pdf/2002.11803.pdf)
>
> A2: Thanks for pointing out the strong baseline. Yes, the Adam(AdamW) baseline in our paper stronger than most reported baselines because we fine-tuned the baseline more carefully. For example, the baseline in our paper outperform the baseline you mentioned by more than two points. Compared to the baselines in other papers, the improvement of AdamS is significant. This might a result of another advantage of AdamS that Adam/AdamW is less robust to the hyperparameters than AdamS. Thus, fine-tuning hyperparameters for Adam is time-consuming and difficult for some papers. This explains why Adam has weaker baseline performance than our in some related papers.
>
> Q3: The connection between section 2 and the rest seems a little weak.
>
> A3: The theoretical analysis in section 2 inspired us to design Rule 2 and Stable Weight Decay. We will try to make the connection more close in the revised version.
>
> Q4: How does eq7 motivate rule 2? rule 1 and rule 2 have the same scaling with respect to eq7. Why does increasing weight decay improve convergence but not increasing the learning rate?
>
> A4: Rule 2, multiplying weight decay by $k$, usually does not break the convergence guarantee of stochastic optimization, while Rule 1, multiplying the learning rate by $k$, may cause bad convergence or even optimization divergence during training. In practice, one definitely cannot use a too large learning rate to achieve good performance. For example, a common convergence guarantee that requires $\eta \leq \frac{C}{\sqrt{t+1}}$ is well known in convergence analysis of stochastic optimization literatures [1-2]. Eq (7) may reflect the effectiveness of large weight decay, and convergence guarantees may prevent using too large learning rates. We will add some discussions and references to clarify this point in the revised version.
>
> References:
>
> [1] Yan, Y., Yang, T., Li, Z., Lin, Q., & Yang, Y. (2018, January). A unified analysis of stochastic momentum methods for deep learning. In IJCAI International Joint Conference on Artificial Intelligence.
>
> [2] Ghadimi, S., & Lan, G. (2013). Stochastic first-and zeroth-order methods for nonconvex stochastic programming. SIAM Journal on Optimization, 23(4), 2341-2368.
>
> [3] Loshchilov, I., & Hutter, F. (2018, September). Decoupled Weight Decay Regularization. In International Conference on Learning Representations.

---

> > ### Comment · Reviewer_1qbp · 2021-11-22
> > **Thanks for the responses.**
> >
> > Thanks for the responses, I will keep my score the same.
> >
> > I had a quick question looking at the addition of table 3, how does this discussion about optimal L2s compare with that of Lewkowycz, Gur-Ari? They seem to have very similar results (see figure 4 for example).

---

> > > ### Author Response · Authors · 2021-11-23
> > > **Discussions on Table 3**
> > >
> > > Thanks for the reviewer's quick feedbacks.
> > >
> > > Figue1(b) given by Lewkowycz, Gur-Ari actually means that the performance of the predicted optimal weight decay and the performance of the observed optimal weight decay are close. However, we may not directly observe the difference between the predicted optimal weight decay and the performance of the observed optimal weight decay.
> > >
> > > While Figure 4 given by Lewkowycz, Gur-Ari presented the direct comparison between the predicted optimal weight decay and the observed optimal weight decay, it is difficult to recognize the order of the magnitude of the difference from Figure 4. We think the log-scale comparison may be more informative here. Because, in practice of fine-tuning weight decay, we mainly need to tune its order of the magnitude.
> > >
> > > Table 3 in our revised version directly supports that the predicted optimal weight decay is fully consistent with the observed optimal weight decay in the sense of the order of the magnitude.

---

### Official Review · Reviewer_6kHA · 2021-10-28

**Correctness:** 4
**Technical Novelty And Significance:** 3
**Empirical Novelty And Significance:** 3
**Recommendation:** 8
**Confidence:** 4

**Main Review:**

The paper introduced the readers on vanilla weight decay, first proposed by Hansan and
Pratt (1989). A more popular version is: θt = (1 − ηtλ)θ_(t−1) − η_t δL(θ_(t−1))/δθ
where η_t is  the learning rate at the step, and the weight decay is coupled with the learning rate scheduler. It is consistently used in several deep learning regularizations. The paper tries to explain some of the not known reasons why weight decay is coupled with the learning rate iteration scheme. It came up with some theoretical proofs on why the above
equation is better than the original version Hanson and Pratt came up with. By citing current research, the paper explained some of the effects of weight decay, such as: Biasing stationary points and accelerating convergence of backpropagation.
Another section looked at the implications of weight decay in larger batch training, which according to research, can efficiently utilize the parallel computation to speed up the training of deep neural networks. It compared the prior work on linearly increasing the
learning rate when batch size increases to linearly increasing the weight decay by the same factor. The latter approach (which the paper novelly proposed) helped in addressing the bad convergence problem that models with high-batch sizes frequently deal with.
Finally, the paper proposed AdamS, a variation of Adam optimizer, with an additional weight decay term in the update rule. It compared its results with other learning algorithms like Adam, AMSGrad, RAdam, and Padam and found that AdamS performed slightly better than
others. In the supplementary sheets, the proofs for stated theorems were given, details on specific models used were provided, and a combination of SWD with other adaptive variants was stated.

Novelty
1. Previous work hadn’t caught on why the linear scaling of learning rate isn’t entirely helpful for higher-batch training data because of low convergence. The current paper came up with a distinct approach that sorted this issue out.
2. Previous work did not focus on the reasons why weight decay should be coupled with the learning rate scheduler, which the current paper explored and tried to exploit with its novel approaches.
3. The paper designed an approach to schedule wait decay that made improvements over L2 Regularization and Decoupled Weight Decay.
4. SWD, the technique proposed by this paper, can be used with a slight modification to the current learning algorithms. It’s not an entirely new algorithm in itself, but it utilizes benefits from the currently popular methods and helps provide a better regularization
and thus better performance.

Additional comments
An insight into future work can be provided on whether there is a plan to target the paper’s limitations, i.e., to come up with a weight-decay scheduler for language models in Natural Language Processing that consistently outperforms L Regularization.

**Summary Of The Paper:**

This paper focuses on Weight decay and its importance in Regularization during training.
The purpose of the paper is to propose an interpretation of weight decay for learning
dynamics, coming up with a mechanism to improve weight decay for larger batch sizes and
developing a schedular called Stable Weight Decay (SWD).


**Summary Of The Review:**

The paper is interesting, free from technical errors, useful and fairly rigorous.

---

> ### Author Response · Authors · 2021-11-22
> **Responses (1) to Reviewer 6kHA**
>
> We thank the reviewer for the strong interest and the kind support to our work.
>
> Your comments on the novelty of our work and the paper structure are highly appreciated. Our work focused on novel properties of weight decay, which is a popular and even necessary regularization technique for training almost all deep models. The importance of weight decay also makes our main contributions very valuable for the deep learning community.
>
> Comment1: Previous work hadn’t caught on why the linear scaling of learning rate isn’t entirely helpful for higher-batch training data because of low convergence. The current paper came up with a distinct approach that sorted this issue out.
>
> A1: Previous works failed to recognize the convergence problem in large-batch training. We pointed out this hidden problem and came up with a novel weight-decay linear scaling rule.
>
> Comment 2: Previous work did not focus on the reasons why weight decay should be coupled with the learning rate scheduler, which the current paper explored and tried to exploit with its novel approaches.
>
> A2: Right, the fact that weight decay should be coupled with the learning rate scheduler is not well known by deep learning researchers. We note that even some popular open-source optimizers also made mistakes here. For example, AdaBound with Decoupled Weight Decay, implemented by the original authors (https://github.com/Luolc/AdaBound), did not adapt the weight decay to the learning scheduler. This improper implementation seriously harm its empirical performance, but not many researchers realized why before. It will be beneficial to spread the fact as “common sense” for the community in future.
>
> Comment3: The paper designed an approach to schedule weight decay that made improvements over L2 Regularization and Decoupled Weight Decay.
>
> A3: Our work is the first to open the possibility and the power of scheduling weight decay. This approach may be promising as a kind novel training method in future. The original decoupled weight decay becomes a special case of our method.
> Comment4: SWD, the technique proposed by this paper, can be used with a slight modification to the current learning algorithms. It’s not an entirely new algorithm in itself, but it utilizes benefits from the currently popular methods and helps provide a better regularization and thus better performance.
>
> A4: Stable Weight Decay is a novel implementation of generalized weight decay. Thus, it can be easily combined with advanced optimization methods. It may further expand the power of regularization and optimization. This is another important advantage of our method.

---

### Official Review · Reviewer_AwnS · 2021-11-03

**Correctness:** 2
**Technical Novelty And Significance:** 2
**Empirical Novelty And Significance:** 2
**Recommendation:** 3
**Confidence:** 4

**Main Review:**

The paper is well written and easy to follow. But I find the interpretation on the effect of weight decay is not convincing, and the experiment results has marginal improvements.

# Major concerns

-  The definition on "stability of the stationary point" is ambiguous. Since there is no re-definition on "stationary point" in this paper, I think the stationary point in this paper still indicates the points whose (full) gradient w.r.t. loss function is zero. Then what is the loss function in this paper? Does it involve $l^2$ regularization part? This should be clarified: If $l^2$ regularization is not involved in, the stationary points w.r.t. the loss function must be fixed whatever optimization method is applied, the definition of stability is redundant; If not, then another issue raises, I will show it below.

- Theorem 1 is trivial. The fact that GD with weight decay as Eq.(1) cannot converge to any stationary points has known long before. However, being unable to converge to a fixed point is not always a disadvantage especially in deep learning tasks. A bunch of work [Hoffer, et al. 2017; Kleinberg, et al. 2018; kunin. et al. 2021] have proven, in either theorem or empirical aspects, that modern neural networks can get better performance by escaping from bad local optimum, or continue moving even after obtaining its best performance. So demonstrating the disadvantage of weight decay formed as Eq.(1) by "stability" is not convincing.

- Assumption 1 is not reasonable in the topic discussed in this paper. On one hand, this paper refer to the results of Li, et al. (2020), Van Laarhoven, 2017, etc,  so I assume the authors have taken batch normalization into account. But the hessian of normalized network is not semi-definite on any well-defined point (except origin point). Beside, the intrinsic learning rate proposed by Li, et al. (2020) is based on a special phenomenon in learning dynamics of normalized neural network, the convergence rate $1/\lambda^*$ indicates the convergence to equilibrium state not the convergence of loss. This is totally different from quadratic cases. More quantitative results about effective learning rate in equilibrium can be seen in a missing reference Wan et al, 2020, in which the roles of weight decay, learning rate, and momentum factor have been thoroughly revealed in learning dynamics of normalized neural network.

On the other hand, approximating the loss landscape as quadratic well is only reasonable when training is close to end [kunin, et al. 2021]. The hessian matrix varies vastly during the whole training procedure, so any insights drawn from the quadratic approximation is not suitable to characterize the whole training procedure, neither does the practical methods.

- Again, theorem 2 is trivial and well known as a result of naive linear regression GD case. I cannot see any significance and novelty from it and its corollaries.

- The reasons why weight decay improves large-batch training are not justified in section 3. First, the results in theorem 1, 2 did not reflect the effect of batch size. They are essentially full batch cases. Some reasons are either not justified, or contrary to existing work:

"Sometimes, the noise magnitude is not the performance bottleneck"

Please justify it. At least in CIFAR and Imagenet tasks, Smith, et al. (2020) and You, et al. (2020) have taken massive computation resource to confirm that with standard training settings, performance degrades due to full batch GD not insufficient training budget.

"If the bad convergence problem is the performance bottleneck, the learning-rate linear scaling rule will even be harmful to
large-batch training due to slower convergence."

Why? Please justify it.

"According to Equation (7), we propose the weight decay linear scaling rule as Rule 2. Because large weight decay may accelerate the convergence"

Why? What's the difference between multiplying learning rate by $k$ and multiplying WD by $k$? Eq.(7) cannot reflect the difference.

- In Adaptive learning rate part (section 4) The intuition is unreasonable:

"We interpret $\eta v_t^{-1/2}$ as the effective learning rate for multiplying the gradients."

Why? Is it reasonable regarding $v_t^{1/2}$ is actually a vector?

" The expected solution learned by AdamW in the longe-time limit is ..." , "In the long-time limit, Equation (10) can indeed fix the stability
problem of the expected solution"

Is it correct? Note the explicit solution of linear regression GD highly relies on the constant value of learning rate and weight decay, it's not reasonable to derive the solution of adaptive optimizer merely by substituting some letters in the formula, rigorous proof is needed.

"...where $\bar{v}_t$ is the mean of all elements of the vector $v_t$..."

Does this relaxation preserve the advantages of the original form of Adaptive weight decay mentioned before? Please justify it.

- Experiment results seems not persuasive. The proposed method are only applied on CIFAR/Imagenet datasets, comparing with Adam/AdamW. It is less meaningful to design a new training method which can only obtain better test performance than Adam while still worse than SGD (with momentum) in CIFAR/Imagenet experiment. The authors can prove its effectiveness in some cases where Adam/AdamW is SOTA, like some transformer tasks (Dosovitskiy, et al. 2020; Liu, et al. 2021).

**Summary Of The Paper:**

In this paper, the authors propose a new interpretation on the effect of weight decay in learning dynamics of neural network. Based on their interpretation, they design a weight-decay linear scaling rule for large-batch training, and a learning-rate-aware scheduler for weight decay in common settings. They verify the effectiveness of the proposed method on CIFAR, Imagenet experiments.

**Summary Of The Review:**

Overall speaking, the assumption used in this paper is too strong to simplify the real cases properly, so the theoretical interpretation established on this assumption is unconvincing. Besides, the large training part lacks justification on some claims, while there seems to be some errors in derivation on the proposed method SWD. The experiment results are not convincing either. The current quality is below the acceptance threshold.

---

> ### Author Response · Authors · 2021-11-22
> **Responses (1) to Reviewer AwnS**
>
> We appreciate the reviewer for the hard work and helpful comments.
>
> The main concerns have been duly addressed below.
>
>
> Q1: What’s the definition of “stationary points”? Is the $L_{2}$ regularization/weight decay involved in the loss function?
>
> A1: Yes, the regularization term should be included into the loss function when we talk about stationary points. If $L_{2}$ regularization is involved, the loss function will be $L(\theta) + \frac{1}{2} \lambda \| \theta\|^{2}$. However, if vanilla weight decay by Eq (1) is involved, the loss function will be $L(\theta) + \frac{\lambda^{\prime}}{\eta} \| \theta\|^{2} $. Obviously, the regularized loss function can be unstable due to scheduled or adaptive learning rates at different steps of training.
>
>
> Q2: Being unable to converge to a fixed point is not always a disadvantage especially in deep learning tasks. A bunch of work [Hoffer, et al. 2017; Kleinberg, et al. 2018; kunin. et al. 2021] have proven, in either theorem or empirical aspects, that modern neural networks can get better performance by escaping from bad local optimum, or continue moving even after obtaining its best performance.
>
> A2: We argue that escaping from bad minima does not mean non-convergence during training. The convergence guarantee is always theoretically important for stochastic optimization, while stochastic optimization methods can escape bad sharp minima and produce the posterior (a sequence of $\theta$). The common convergence measure of stochastic optimizers is that the minimum expected gradient norm may approach to zero during training, such as $E[\| \nabla L(\theta_{k})\|^{2}] \leq \frac{C}{\sqrt{t+1}} $ [11-12]. All widely used optimizers, including SGD, Momentum, and Adam, have clear convergence guarantees.
>
> Optimizers with no convergence guarantees may cause undesirable training behaviors and may not be popular in practice. For example, non-convergence may not minimize the training loss well, and, thus, the trained model have bad performance on both the training dataset and the test dataset [18]. Thus, optimizing a loss with no convergence guarantee can be a significant disadvantage. This conclusion is also supported by our experimental results.
>
> Q3: Assumption 1 is not reasonable in the topic discussed in this paper. Any insights drawn from the quadratic approximation is not suitable to characterize the whole training procedure, neither does the practical methods.
>
> A3: Assumption 1 is very common in the papers on optimization dynamics of deep learning [1-5], as long as the theoretical analysis provides insights to deep learning.
>
> We argue that, while Assumption 1 (the quadratic approximation) limits the application scope, our theoretical analysis under Assumption 1 still can provide novel insights. “All models are wrong, but some are useful.” The insights drawn from the quadratic approximation is of course not completely right to characterize the whole training procedure. But it can help provides useful insights beyond existing papers. It is known that stochastic optimization can escape multiple bad minima [14-16] and finally converge to a good minimum during training. Thus, quadratic approximations can be repeatedly applied near each minimum during training. Theorem 2 is a useful tool to understand convergence behaviors which may repeatedly happen during searching minima of deep learning.
> Even if the quadratic approximation is not always mild, the theoretical analysis may still inspire practical algorithms. The experimental results support our method.
>
>
> Q4: The intrinsic learning rate proposed by Li et al. (2020) is based on a special phenomenon in learning dynamics of normalized neural network, the convergence rate 1/λ∗ indicates the convergence to equilibrium state not the convergence of loss. This is totally different from quadratic cases.
>
> A4: Yes, the setting in Li et al. (2020) is different from quadratic cases. However, this does not change the conclusion. Our theoretical analysis is independent of the scale invariant property of deep loss landscape. Our finding is that, with or without normalization layers, the conclusion that the learning rate and weight decay are linearly coupled still holds in our theoretical analysis. In the empirical analysis, we may also observe that Rule 2 holds well in the experiments of VGG, which is not scale invariant. Thus, our contribution is novel and beyond existing papers. We will add more discussions for clarifying the point in the first paragraph of Section 6 and the last paragraph of Appendix C.
>
>
> Please refer to Responses (2) for more discussions.

---

> ### Author Response · Authors · 2021-11-22
> **Responses (2) to Reviewer AwnS**
>
> Q5: Theorem 2 is trivial and well known as a result of naive linear regression GD case.
>
> A5: We agree that we can obtain a similar result like Theorem 2 in some specific case like linear regression GD. But this does not harm the contributions of our work(Please See Page 2). For example, we are the first to use Theorem 2 to derive the novel Rule 2 for large-batch training of deep networks. Rule 2 is a novel and significant finding, which is beyond existing well-known Rule 1. It is good that simple theorems help understand novel insights. We are glad that we can derive novel and useful insights for deep learning from simple theoretical analysis.
>
> Q6: The reasons why weight decay improves large-batch training are not justified in section 3. The results in theorem 1, 2 did not reflect the effect of batch size.
>
> A6: We has clearly noted that weight decay may accelerate convergence (Please see Effect 2 in Section 2), while bad convergence is the performance bottleneck of large-batch training (Please see Page6). Given the number of training epochs, if we multiply the batch size by $k$, the number of training iterations will be divided by $k$. Thus, the number of training iterations may be much less than the expected number of iterations for good convergence discussed in Section 2. In this case, Rule 2 works well, theoretically and empirically.
>
> We totally agree that the results in Theorems 1 and 2 did not reflect the effect of batch size. This is why the theorems can well describe large-batch training dynamics. Our theoretical analysis, including Theorems 1 and 2, focused on deterministic optimization dynamics, because the deterministic component alone has already supported the empirical results. It means our theoretical analysis has captured the main theoretical mechanism of the reported empirical results.
>
> Q7: Please justify "Sometimes, the noise magnitude is not the performance bottleneck".
>
> A7: Our experimental results in Figure 3 may justify this sentence. If the noise magnitude is the performance bottleneck, we may apply a large learning rate to improve large-batch training as the conventional Rule 1 suggests. Because the gradient noise strength in optimization dynamics is proportional to the ratio of learning rate to batch size, namely as $\frac{\eta}{B}$, which is a famous result in optimization dynamics literatures [2,6-10]. Figure 3 presents the result of a common setting where even slightly increasing learning can harm the performance of large-batch training.
>
> Q8: Please justify "If the bad convergence problem is the performance bottleneck, the learning-rate linear scaling rule will even be harmful to large-batch training due to slower convergence."
>
> A8: Our experimental results in Figure 3 may also justify this sentence. Because convergence errors directly relates to the learning rate in stochastic optimization. Using a too large learning rate may cause bad convergence or even optimization divergence during training. For example, a common convergence guarantee that requires $\eta \leq \frac{C}{\sqrt{t+1}}$ is well known in convergence analysis of stochastic optimization literatures [2, 11-12]. Thus, we must have small enough $\eta$ for good convergence in the final training phase, where the number of iterations $t$ becomes large. We will clarify this point more clearly in the revised version.
>
> Q9: What's the difference between multiplying learning rate by k and multiplying WD by k? Eq.(7) cannot reflect the difference.
>
> A9: Rule 2, multiplying weight decay by $k$, usually does not break the convergence guarantee of stochastic optimization, while Rule 1, multiplying the learning rate by $k$, may cause bad convergence or even optimization divergence during training. In practice, one definitely cannot use a too large learning rate to achieve good performance. For example, a common convergence guarantee that requires $\eta \leq \frac{C}{\sqrt{t+1}}$ is well known in convergence analysis of stochastic optimization literatures [11-12]. Eq (7) may reflect the effectiveness of large weight decay, and convergence guarantees may prevent using too large learning rates. We will add some discussions and references to clarify this point in the revised version.
>
> Q10: "We interpret $\eta v_{t}^{-\frac{1}{2}}$ as the effective learning rate for multiplying the gradients." Is it reasonable regarding $ v_{t}^{-\frac{1}{2}}$ is actually a vector?
>
> A10: It is reasonable if we consider the effective learning rate as a vector for Adam. And the vector $\eta v_{t}^{-\frac{1}{2}}$ exactly reflects the effective learning rates for each dimension element-wisely.
>
> Please refer to Responses (3) for more discussions.

---

> ### Author Response · Authors · 2021-11-22
> **Responses (3) to Reviewer AwnS**
>
> Q11: " The expected solution learned by AdamW in the longe-time limit is Eq (9)" , "In the long-time limit, Equation (10) can indeed fix the stability problem of the expected solution" Is it correct?
>
> A11: We are sorry for the confusing expressions. We actually mean that, when we consider the true loss function optimized by Adam at step-$t$ (given $v_{t}$), the minimum of the true loss function at step-$t$ is given by Eq (9). It demonstrates that the true loss function optimized by Adam has no stable minimum at different steps. Eq (10) can fix the stability problem because the minimum of the true loss function optimized by Adam at step-$t$ is stable at different steps. We will revise the manuscript to make our point more clear.
>
> Q12: " Eq (11) where $v_{t}$ is the mean of all elements of the vector $v_{t}$..." Does this relaxation preserve the advantages of the original form of Adaptive weight decay mentioned before? Please justify it.
>
> A12: As we discussed above Eq (10), there is no ideal solution to fix the stability problem and the convergence problem at the same time for adaptive gradient methods. Our method aims at providing a way to balance the tradeoff between the stability and the convergence in presence of Adaptive Learning Rate. The benefit of balancing tradeoff can be verified by the experimental results, such as Table 1. This is also inspired by the finding that weight decay should be adapted to the learning rate in Section.
>
> Q13: Experiment results seems not persuasive. The proposed method are only applied on CIFAR/Imagenet datasets, comparing with Adam/AdamW. It is less meaningful to design a new training method which can only obtain better test performance than Adam while still worse than SGD (with momentum) in CIFAR/Imagenet experiment. The authors can prove its effectiveness in some cases where Adam/AdamW is SOTA, like some transformer tasks (Dosovitskiy, et al. 2020; Liu, et al. 2021).
>
> A13: First, we note that, in the experiments of large-batch training where Adam outperforms SGD, the proposed AdamS can still show the performance improvement over Adam with $L_{2}$ regularization and Adam with decoupled weight decay. For example, the test accuracy of Adam (90%) may significantly outperform SGD (87%) by three points for ResNet18 in our baseline large-batch experiments (B=16384). It is known that Adam/AdamW is the popular optimizer with convergence guarantees for large-batch training. Our experimental results in Figure 9 demonstrated AdamS outperforms Adam and AdamW on large-batch training of ResNet18. In the revised version, we also supplied the large-batch training experiments of VGG16, which also verified the advantage of SWD over $L_{2}$ regularization and decoupled weight decay.
>
> Second, as Figures 7 and 8 show, AdamS is more robust to the training hyperparameters than Adam and AdamW. This proves another practical advantage of SWD over $L_{2}$ regularization and decoupled weight decay for Adam, because tuning hyperparameters can be less time-consuming and difficult for AdamS.
>
> Third, Figure 17 further supports that AdamS converges better (to lower training losses) than both AdamW and Adam. A similar empirical evidence is used by [17] to prove the advantage of AdamW over Adam with $L_{2}$ regularization.

---

> ### Author Response · Authors · 2021-11-22
> **References for Reviewer AwnS**
>
> References:
>
> [1] Zhang, G., Li, L., Nado, Z., Martens, J., Sachdeva, S., Dahl, G., ... & Grosse, R. B. (2019). Which algorithmic choices matter at which batch sizes? insights from a noisy quadratic model. Advances in neural information processing systems, 32, 8196-8207.
>
> [2] Xie, Z., Sato, I., & Sugiyama, M. (2020, September). A Diffusion Theory For Deep Learning Dynamics: Stochastic Gradient Descent Exponentially Favors Flat Minima. In International Conference on Learning Representations.
>
> [3] Mandt, S., Hoffman, M. D., & Blei, D. M. (2017). Stochastic Gradient Descent as Approximate Bayesian Inference. Journal of Machine Learning Research, 18, 1-35.
>
> [4] Li, Q., Tai, C., & Weinan, E. (2017, July). Stochastic modified equations and adaptive stochastic gradient algorithms. In International Conference on Machine Learning (pp. 2101-2110). PMLR.
>
> [5] Zhou, P., Feng, J., Ma, C., Xiong, C., & Hoi, S. C. H. (2020). Towards Theoretically Understanding Why Sgd Generalizes Better Than Adam in Deep Learning. Advances in Neural Information Processing Systems, 33.
>
> [6] Wu, J., Hu, W., Xiong, H., Huan, J., Braverman, V., & Zhu, Z. (2020, November). On the noisy gradient descent that generalizes as sgd. In International Conference on Machine Learning (pp. 10367-10376). PMLR.
>
> [7] Jastrzębski, S., Kenton, Z., Arpit, D., Ballas, N., Fischer, A., Bengio, Y., & Storkey, A. (2017). Three factors influencing minima in sgd. arXiv preprint arXiv:1711.04623.
>
> [8] Mandt, S., Hoffman, M. D., & Blei, D. M. (2017). Stochastic Gradient Descent as Approximate Bayesian Inference. Journal of Machine Learning Research, 18, 1-35.
>
> [9] He, F., Liu, T., & Tao, D. (2019). Control batch size and learning rate to generalize well: Theoretical and empirical evidence. Advances in Neural Information Processing Systems, 32, 1143-1152.
>
> [10] Keskar, N. S., Mudigere, D., Nocedal, J., Smelyanskiy, M., & Tang, P. T. P. (2016). On Large-Batch Training for Deep Learning: Generalization Gap and Sharp Minima. In International Conference on Learning Representations.
>
> [11] Yan, Y., Yang, T., Li, Z., Lin, Q., & Yang, Y. (2018, January). A unified analysis of stochastic momentum methods for deep learning. In IJCAI International Joint Conference on Artificial Intelligence.
>
> [12] Ghadimi, S., & Lan, G. (2013). Stochastic first-and zeroth-order methods for nonconvex stochastic programming. SIAM Journal on Optimization, 23(4), 2341-2368.
>
> [13] Goyal, P., Dollár, P., Girshick, R., Noordhuis, P., Wesolowski, L., Kyrola, A., ... & He, K. (2017). Accurate, large minibatch sgd: Training imagenet in 1 hour. arXiv preprint arXiv:1706.02677.
>
> [14] Kleinberg, B., Li, Y., & Yuan, Y. (2018, July). An alternative view: When does SGD escape local minima?. In International Conference on Machine Learning (pp. 2698-2707). PMLR.
>
> [15] Zhu, Z., Wu, J., Yu, B., Wu, L., & Ma, J. (2019, May). The Anisotropic Noise in Stochastic Gradient Descent: Its Behavior of Escaping from Sharp Minima and Regularization Effects. In International Conference on Machine Learning (pp. 7654-7663). PMLR.
>
> [16] Jastrzębski, S., Kenton, Z., Arpit, D., Ballas, N., Fischer, A., Bengio, Y., & Storkey, A. (2017). Three factors influencing minima in sgd. arXiv preprint arXiv:1711.04623.
>
> [17] Loshchilov, I., & Hutter, F. (2018, September). Decoupled Weight Decay Regularization. In International Conference on Learning Representations.
>
> [18] Reddi, S. J., Kale, S., & Kumar, S. (2018, February). On the Convergence of Adam and Beyond. In International Conference on Learning Representations.

---

### Author Response · Authors · 2021-11-22
**General Responses**

We sincerely appreciate all reviewers for the hard work and helpful comments.

We would like to address all reviewers’ main concerns in the corresponding responses.

We also updated our manuscript according to the comments.

The change we made mainly includes:
-	We explain why our results are independent of the scale-invariant property of the loss function. We presented the empirical results of VGG16 (which has no BatchNorm) in Figure 16 to support this point.
-	We revised some discussions and statements to improve presentation.
-	We present novel empirical results in Table 3 to better support that the optimal weight decay is approximately inverse to the number of epochs.

---

### Decision · Program_Chairs · 2022-01-20

**Decision:**

Reject

**Comment:**

This paper analyzes the effects of the weight decay hyperparameter, and based on this analysis, proposes methods to schedule the weight decay. Overall, while I'm glad that more work is being done on understanding the effects of weight decay, I don't think this submission is of sufficient quality for ICLR.

Theorem 1 is simply re-expressing the well-known fact that if the regularization version of weight decay is used, then (simply because it's based on a single objective function) the stationary points are invariant to the choice of learning rate. This may not be apparent due to the misused terminology: "invariant" is referred to as "stable", but "stable stationary point" has a technical meaning very different from the one used here.

Corollary 2 essentially shows that the optimum of the regularized loss is different from the optimum of the unregularized loss. The authors conclude from this that the optimal value of lambda is 0 from the perspective of test error, which is unwarranted.

Overall, the paper centers around the interaction between learning rates and the weight decay parameter. However, as various reviewers point out, this interaction has been analyzed in detail for networks with normalization layers, and normalization completely changes the nature of the interaction. So any analysis would either need to take this into account or limit the scope to networks without normalization.

I encourage the authors to take the reviewers' feedback into account and improve the paper for the next submission cycle.